# Tamoxifen therapy in a murine model of myotubular myopathy

Nika Maani[1,2], Nesrin Sabha[1,2,3], Kamran Rezai [1,2], Arun Ramani[1,4,5], Linda Groom[6], Nadine Eltayeb[1], Faranak Mavandadnejad[1], Andrea Pang[1], Giulia Russo [7], Michael Brudno[1,4,5], Volker Haucke[7], Robert T. Dirksen[6] & James J. Dowling[1,2,3]

Myotubular myopathy (MTM) is a severe X-linked disease without existing therapies. Here, we show that tamoxifen ameliorates MTM-related histopathological and functional abnormalities in mice, and nearly doubles survival. The beneficial effects of tamoxifen are mediated primarily via estrogen receptor signaling, as demonstrated through in vitro studies and in vivo phenotypic rescue with estradiol. RNA sequencing and protein expression analyses revealed that rescue is mediated in part through post-transcriptional reduction of dynamin-2, a known MTM modifier. These findings demonstrate an unexpected ability of tamoxifen to improve the murine MTM phenotype, providing preclinical evidence to support clinical translation.

---

[1] Program for Genetics and Genome Biology, Hospital for Sick Children, 686 Bay Street, Toronto, ON CAN M5G 0A4, Canada. [2] Department of Molecular Genetics, University of Toronto, Medical Science Building, Room 4386, 1 King's College Cir, Toronto, ON CAN M5S 1A8, Canada. [3] Department of Paediatrics, University of Toronto, Room 1436D, 555 University Avenue, Toronto, ON CAN M5G 1X8, Canada. [4] Department of Computer Science, University of Toronto, Pratt Building Room 286C, 6 King's College Rd, Toronto, ON CAN M5S 3G4, Canada. [5] Centre for Computational Medicine, The Hospital for Sick Children, 686 Bay Street, Toronto, ON CAN M5G 0A4, Canada. [6] Department of Pharmacology and Physiology, University of Rochester Medical Center School of Medicine and Dentistry, 601 Elmwood Ave, Box 711, Rochester, NY 14642, USA. [7] Department of Molecular Pharmacology and Cell Biology, Leibniz-Forschungsinstitut für Molekulare Pharmakologie (FMP), 13125 Berlin, Germany. These authors contributed equally: Nika Maani, Nesrin Sabha. Correspondence and requests for materials should be addressed to J.J.D. (email: james.dowling@sickkids.ca)

Myotubular myopathy (XLMTM or MTM) is a childhood muscle disease characterized by profound, neonatal onset weakness, severe disabilities (including wheelchair and ventilator dependence), and early death[1]. It is traditionally defined by characteristic muscle pathology that includes increased central nuclei, disorganized perinuclear organelles, and myofiber hypotrophy[2]. MTM is a X-linked condition, primarily affecting boys, that is caused by loss-of-function mutations in the myotubularin (*MTM1*) gene[3]. *MTM1* encodes a lipid phosphatase that regulates the levels of 3-position phosphoinositides and participates in endosomal sorting[4].

At present, there are no approved therapies for MTM. Based on previous pre-clinical studies[5,6], pyridostigmine is used off label in MTM patients with modest effect. *MTM1* gene therapy, which has shown great promise in mouse and dog models of the disease[7], has just entered human clinical trials (NCT03199469). Additional potential therapeutic approaches with efficacy in an MTM mouse model are dynamin-2 (DNM2) downregulation and inhibition of the phosphoinositide 3-kinase PIK3C2B[8–10]. However, outside of gene therapy, no small molecule or non-gene-based therapy is close to entering the clinical arena; given the severe nature of this disease, there thus remains a critical need to develop new and effective treatment strategies.

As a baseline control for Cre-mediated removal of *Pik3c2b* in the *Mtm1* knockout (or MTM) mouse model[8], we treated non-Cre, non-floxed MTM mice with a short course of tamoxifen and noted a slight, non-significant, right shift in survival[8]. This led us to hypothesize that chronic tamoxifen therapy can independently improve the MTM phenotype. We test this using daily tamoxifen treatment and observe improved survival as well as significant restoration of muscle structure and function. We find that the beneficial effects of tamoxifen treatment are primarily mediated through non-nuclear estrogen receptor alpha dependent effect(s), including a reduction of dynamin-2 protein levels. In total, our results show that tamoxifen treatment ameliorates murine MTM, and, because tamoxifen is a commonly used, FDA approved compound, can be considered for rapid clinical translation.

## Results

**Tamoxifen treatment increases survival of *Mtm1* knockout mice.** The MTM mouse model (abbreviated as *Mtm1* KO or MTM) used for this study was generated by removal of *Mtm1* exon 4 by homologous recombination[11]. It faithfully recapitulates the molecular and histopathologic changes observed in human MTM muscle, and has a severe phenotype that mirrors the human disease. Untreated *Mtm1* KO mice exhibit overt symptoms beginning at ~21 days post birth, develop marked hindlimb weakness at 30 days, and die at a median age of 39 days (longest survivor = 52 days)[8,11]. We treated MTM mice with daily tamoxifen (formulated in chow) or placebo (standard chow). Tamoxifen (TAM) was given at either a low (3 mg/kg) or high (40 mg/kg) dose, with low-dose modeling human pediatric dosing and high dose matching what is used for Cre-lox recombination in mice[12,13]. Tamoxifen exposure was verified by measuring levels of tamoxifen and its metabolites in blood and muscle at 35 days of age (Supplementary Table 1).

We first tested the effect of tamoxifen treatment when started at 21 days of age, a time point at which MTM mice have normal strength but are distinguishable by decreased body weight. When started at this age, both low and high-dose therapy significantly extended the lifespan of *Mtm1* KO mice, with median survival observed at 57 days and 48 days, respectively (Fig. 1a, b).

We next examined tamoxifen therapy starting at 14 days of age, a time point before overt symptoms develop in the *Mtm1* KO mice. To accomplish this, we used low-dose tamoxifen only, as

chronic high-dose exposure is not tolerated by pre-weaned mice and also interferes with maternal breastfeeding. Treatment starting at 14 days significantly prolonged survival as compared to untreated MTM mice or to MTM mice started on tamoxifen at 21 days, with a median survival of 71 days (longest survival = 83 days) (Fig. 1b).

We lastly studied the impact on survival of tamoxifen therapy started after MTM mice exhibit weakness (Fig. 1a, b). To accomplish this, we began tamoxifen treatment at 30 days, a time point we previously established as when hindlimb weakness is fully penetrant[8]. Low-dose therapy provided a significant increase in lifespan (median = 54 days, longest survivor = 56, $p = 0.004$ vs. untreated KO, log-rank (Mantel-Cox) test). High-dose treatment did not extend survival in the composite (median = 35 days), though 3 KOs in this cohort died in the first 2–3 days of therapy onset. However, when these short-lived mice are excluded from analysis, there was a significant increase seen with high-dose tamoxifen as well (median survival = 66 days, longest survivor = 77, $p = 0.025$, log-rank (Mantel-Cox) test). Tamoxifen treatment, therefore, results in a survival benefit even in advanced stages of the murine MTM disease process.

**WT and *Mtm1* KO weights are reduced by high-dose tamoxifen.** As noted above, body weight in untreated *Mtm1* KOs is significantly reduced as compared to wild type (Supplementary Figure 1). We measured daily weights to see whether tamoxifen therapy altered this. Animals treated with low-dose tamoxifen had body weights unchanged as compared to untreated *Mtm1* KOs. Both wild types and *Mtm1* KOs treated with high-dose tamoxifen, however, exhibited decreased weights that were significantly lower than placebo exposed animals after 7 days of treatment (Supplementary Figure 1). This decrease is in keeping with previous studies showing that high-dose tamoxifen is associated with reduced food intake and an initial >10% weight loss[14]. This may explain why high-dose therapy in general does not improve survival to the same extent as low-dose treatment.

We also considered whether weight reduction was the cause for early death in the subset of MTM animals that died within 2–3 days of starting high-dose tamoxifen at 30 days of age. To investigate this, we performed full necropsy on two mice that died precipitously after high-dose therapy, and compared the results to untreated MTM mice, MTM mice treated starting at 21 days, longer lived MTM mice started on tamoxifen at 30 days, and wild types under the same conditions (Supplementary Figure 2). We looked by histopathological analysis, and did not see obvious differences between untreated MTM mice and mice treated under the various conditions in terms of liver, kidney, and heart pathology. However, we did observe severe pulmonary hemorrhage in the 30-day-treated MTM mice with precipitant death. This was not seen with any other condition, and thus suggests that high-dose treatment started on very late stage MTM mice can infrequently promote pulmonary injury.

**Tamoxifen improves muscle function of *Mtm1* KO mice.** We next tested whether tamoxifen altered muscle structure (Figs. 2, 3) and function (Figs. 1, 3) of MTM mice. MTM mice treated with tamoxifen starting at 14 or 21 days qualitatively appeared stronger and more active, as depicted by photomicrograph (Fig. 1c, note improved hindlimb positioning, and Fig. 3a) and by video (Supplementary videos 1–5, note increased activity and climbing behavior). To evaluate muscle function quantitatively, we measured hindlimb grip strength at 35 days of age, an age where we previously established a significant difference between wild type and MTM mice[8]. Remarkably, both low and high-dose tamoxifen treatment of *Mtm1* KO mice restored grip strength to

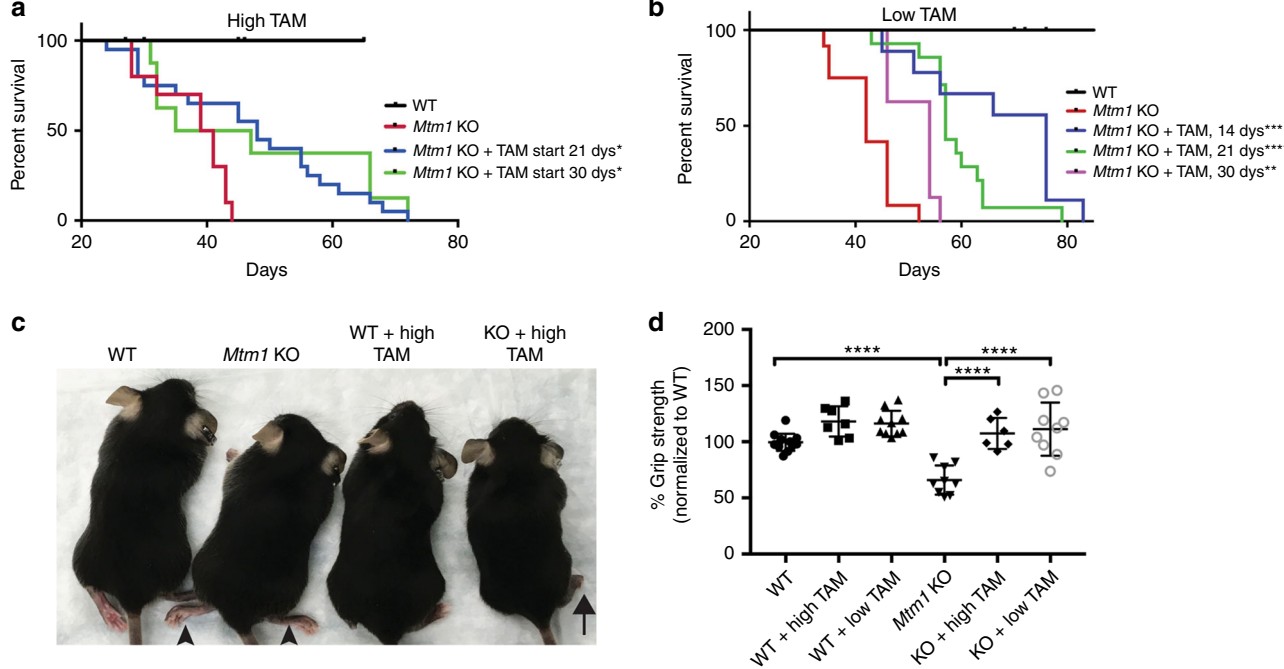

**Fig. 1** Tamoxifen treatment prolongs survival and improves muscle strength in *Mtm1* knockout mice. **a** High-dose Tamoxifen (TAM) treatment starting at 21 days promotes significant improvement in *Mtm1* KO survival [median survival 48 days ($n = 20$) vs. 39 days for untreated KOs ($n = 7$), $*p = 0.05$]. High-dose treatment at 30 days does not improve survival in aggregate but does in the subset of treated animals surviving beyond the first 2 days of treatment [subset median survival 66 days ($n = 5$) vs. 41 days for untreated KOs ($n = 3$), $*p = 0.03$]. **b** Low-dose TAM treatment starting 14, 21, or 30 days significantly improves *Mtm1* KO survival. Low-dose TAM treatment starting 14 days has a median survival of 71 days ($n = 9$) vs. 42 days for untreated KOs, $***p = 0.001$. Low-dose TAM treatment starting 21 days median survival is 57 days ($n = 14$) vs. 42 days for untreated KOs ($n = 5$), $****p = 3.1 \times 10^{-5}$. Low-dose TAM treatment starting 30 days has a median survival of 54 days ($n = 8$) vs. 44 days for untreated KOs ($n = 4$), $**p = 0.004$ **c**. Photomicrograph of WT, *Mtm1* KO, and high-dose TAM treated WT and KOs. Untreated KOs have splayed hind limbs (arrowhead indicates limb weakness). High-dose TAM treated KOs have normal limb positioning and posture. **d** Grip strength was measured at 35 days of age. High-dose TAM treated KOs ($n = 6$) have a mean grip strength of 107 ± 8%, vs. 65 ± 8% for untreated KOs ($n = 4$), $****p < 0.0001$; high-dose TAM treated WT having 118 ± 7% ($n = 7$) vs. 100 ± 8% ($n = 7$) for untreated WT. Low-dose TAM KOs ($n = 9$) mean grip strength = 112 ± 8.9%, vs. 66 ± 10% for untreated KOs ($n = 5$), $****p < 0.0001$, while treated WT = 116 ± 9% ($n = 10$), vs. 100 ± 10% for untreated WT ($n = 6$). All values normalized to untreated WT. Statistical analysis by log-rank (Mantel–Cox) test, two-way ANOVA (Tukey's multiple comparisons/Fisher's least common differences post-test) or unpaired Student's *t*-test

levels indistinguishable from that of wild-type littermates (Fig. 1d). In addition, treatment starting at 14 days preserved grip strength at 50 days of age, a time point where essentially all untreated MTM mice are dead (Fig. 3c).

**Tamoxifen improves muscle structure of *Mtm1* KO mice.** Light and electron microscopy were used to assess skeletal muscle structure (Figs. 2, 3d–h and 4). MTM muscle, in both the mouse model and in patients, is characterized by small myofibers with increased centrally located nuclei, mislocalization of intracellular organelles, and disorganization of sarco-tubular membranes[2]. Using standard histological stains (hematoxylin/eosin and succinate dehydrogenase) and immunofluorescence evaluation of dysferlin localization (used as a marker of sarco-tubular structures[15]), we observed improvements in all relevant aspects of MTM muscle pathology from *Mtm1* KO following high-dose tamoxifen treatment: (1) decreased % central nuclei, (2) increased myofiber size, (3) reduced perinuclear accumulation of oxidative stained organelles, and (4) improved sarco-tubular organization (Fig. 2). Low-dose treatment, as demonstrated with dysferlin immunostaining, likewise improved tubular membrane organization. When started at 21 days, it did not result in improvement in fiber size or central nucleation. When started at 14 days, it restored myofiber size to normal but did not reduce the percentage of central nuclei (Fig. 3e–h).

We also measured fiber type distribution using myosin heavy chain immunostaining (Supplementary Figure 3). While human MTM muscle is characterized by a predominance of type 1 fibers[16], we did not observe any difference in distribution between wild type and untreated *Mtm1* KO mice. With tamoxifen treatment, however, there was a significant increase in Type 2B fibers, with a corresponding reduction in Type 1 and 2A fibers. This was observed with both low and high-dose therapy. This shift may reflect improvements in the EC coupling apparatus and in intracellular calcium handling in tamoxifen treated MTM mice (see below)[17].

**TAM improves triad structure and EC coupling in *Mtm1* KO mice.** Abnormalities in the skeletal muscle triad, a structure that mediates excitation-contraction (EC) coupling[18], are considered one of the most functionally relevant structural changes in MTM muscle[19,20]. Therefore, we determined the impact of tamoxifen treatment on triad structure and function. We examined triad structure by electron microscopy, which has previously been reported as abnormal and reduced in number in *Mtm1* KO mice[8,21]. Remarkably, high-dose tamoxifen treatment restored triad appearance and number in muscle from MTM mice to levels comparable to those of muscle from wild-type mice (Fig. 4a, b). To assess triad function, we measured electrically evoked calcium release in single flexor digitorum brevis muscle fibers isolated from wild type and *Mtm1* KO mice. Untreated MTM

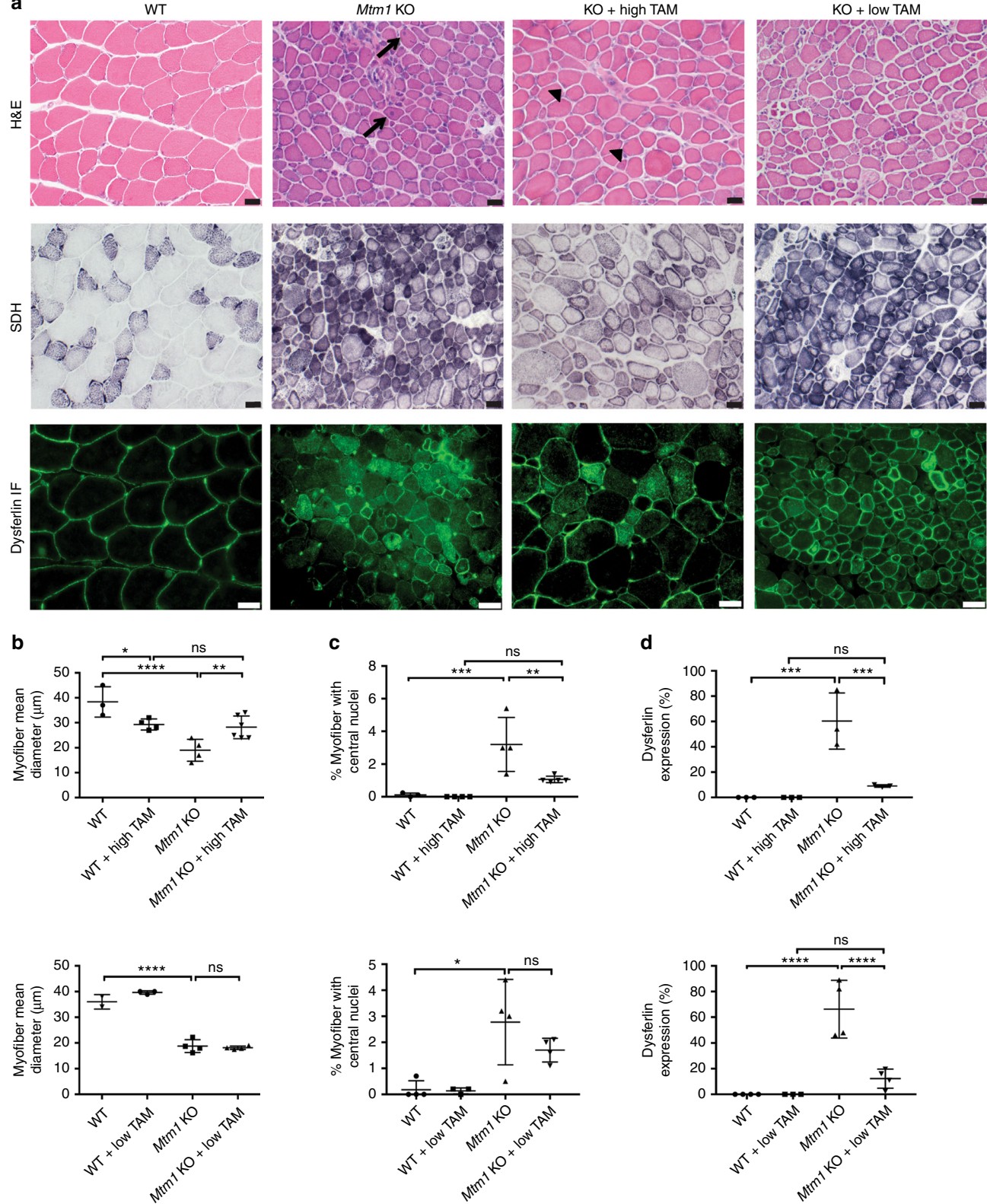

mice exhibited significantly reduced maximal calcium release during both twitch and tetanic stimulation compared to fibers from wild-type littermates. Importantly, high-dose tamoxifen treatment significantly increased maximal twitch and tetanic calcium release in fibers from MTM mice (Fig. 4c–e), consistent with the observed normalization of triad number. While there was a trend toward increase, particularly in the animals started at 14 days (Fig. 3g, h), low-dose tamoxifen treatment did not significantly improve triad number (Fig. 4b). Taken together with the histopathological analyses, these data indicate that tamoxifen results in a dose dependent rescue of key skeletal muscle structures and functions that are defective in untreated MTM mice.

**Fig. 2** Tamoxifen treatment improves *Mtm1* knockout mouse muscle structure. **a** Tibialis anterior muscle stained with hematoxylin and eosin (H&E) and succinate dehydrogenase (SDH), and by anti-dysferlin immunofluorescence (IF). Untreated *Mtm1* KO muscle has centrally located nuclei (arrows in H&E), mitochondrial aggregation on SDH staining, and abnormal distribution of dysferlin staining. High-dose TAM treatment in KOs results in: (1) reduction of central nuclei (arrowheads); (2) improvement in myofiber size; (3) resolution of mitochondrial aggregation; and (4) restoration of dysferlin to the sarcolemmal membrane. Low-dose TAM improves dysferlin localization but does not resolve central nucleation and myofiber hypotrophy. All drug treatments started at 21 days, and histopathology performed at 36 days of age. **b** High-dose TAM treatment increases myofiber size of *Mtm1* KOs. TAM treated WT ($29 \pm 3$ μm, $n = 4$) vs. untreated *Mtm1* KO ($19 \pm 3$ μm, $n = 4$, ****$p < 0.0001$ vs. WT), vs. TAM treated KO ($28 \pm 3$ μm, $n = 6$, **$p = 0.0065$ vs. untreated KO, not significant vs. WT + TAM). **c** High-dose TAM treatment reduces % central nuclei in *Mtm1* KOs (per 100 fibers): TAM treated WT (0%, $n = 4$) vs. untreated *Mtm1* KOs ($3 \pm 0.8\%$, $n = 4$, ***$p = 0.0004$ vs. WT), vs. TAM treated KOs ($1.1 \pm 0.09\%$, $n = 5$, **$p = 0.0024$ vs. KOs and n.s. vs. WT + TAM). **d** High and low-dose TAM treatment restores dysferlin localization to cell membrane. Untreated *Mtm1* KO = $60 \pm 13\%$ cytoplasmic ($n = 3$, ***$p = 0.0002$ vs. WT), KO + high TAM = $9 \pm 1\%$ cytoplasmic ($n = 3$, ***$p = 0.0005$ vs. KO, ns compared to WT + TAM). For low-dose TAM, *Mtm1* KO = $66 \pm 11\%$ cytoplasmic, ($n = 3$, ****$p = < 0.0001$ vs. WT) vs. KO + low TAM = $12 \pm 4\%$ cytoplasmic ($n = 4$, ****$p < 0.0001$ vs. KO, ns compared to WT + TAM). All data points for **b**–**d** presented in Supplementary Table 2. Statistical analyses were conducted by two-way ANOVA, followed by Tukey's multiple comparisons post-test or Fisher's Least Common Differences post-test. For direct two sample comparison, unpaired, parametric two-tailed Student's *t*-test was performed. Scale bars = 20 μm

**Estradiol improves survival of *Mtm1* KO mice**. Tamoxifen is a selective estrogen receptor modifier that, depending on tissue and cellular context, can act either as an agonist or an antagonist[22]. To help distinguish these possibilities, we treated MTM mice with estradiol, a pure estrogen receptor agonist, or fulvestrant, a pure estrogen receptor antagonist (Fig. 5). Daily treatment with estradiol (via subcutaneous injection) starting at 21 days resulted in a significant improvement in survival to a median age of 56 days (Fig. 5a). Fulvestrant, on the other hand, did not improve survival, and MTM animals exposed to this drug trended toward decreased lifespan (Fig. 5b). Estradiol (but not fulvestrant) treated mice also appeared more active and subjectively stronger than placebo treated *Mtm1* KOs (Supplementary videos 6, 7). However, as determined by grip strength analysis and histopathological assessment, we did not detect quantitative improvements in MTM mouse muscle function or parameters of muscle structure with either estradiol or fulvestrant (Fig. 5d–i). These data indicate that tamoxifen, at least in part, is acting via estrogen agonism to promote MTM survival.

**Tamoxifen does not alter the transcriptome of MTM muscle**. The main hypothesized mechanism of action of tamoxifen is transcriptional modulation via estrogen receptor[23,24]. We first determined the protein levels of estrogen receptor isoforms alpha (ERα) and beta (ERβ) in muscle from *Mtm1* KOs. We detected by western blot a significant increase in MTM mice in ERα protein levels (Fig. 6a–f), but no increase (and overall minimal amount of expression) in ERβ (Supplementary Figure 4). Interestingly, both low and high-dose tamoxifen, as well as estradiol, reduced the amount of ERα protein, though expression remained increased compared to wild types (Fig. 6a–f and Supplementary Table 4).

To assess the impact on transcription of tamoxifen therapy, we performed total RNA sequencing from quadriceps muscle (Fig. 6g–h and Supplementary Data 1). In MTM mice without treatment, we observed 849 differentially expressed independent transcripts (Fig. 6h, $p \le 0.01$, $n = 3$ mice per group for WT and *Mtm1* KO, and 4 mice per group for WT + TAM and *Mtm1* KO + TAM, pairwise comparisons by DESeq and edgeR R/Bioconductor packages). The most highly enriched GO term groupings included general muscle terms (Z disc, costamere, sarcomere, and sarcolemma), and organelle terms such as extracellular exosomes and sarcoplasmic reticulum (Supplementary Data 2). Surprisingly, and despite the obvious phenotypic correction promoted by tamoxifen, there was very little change in the transcriptome of the MTM mouse muscle after high-dose tamoxifen treatment, with only 29 transcripts showing significant differential expression ($p \le 0.01$) (Fig. 6h). Furthermore, direct examination of known estrogen receptor response genes in MTM

mice showed no change with tamoxifen exposure (Fig. 6i). This includes ERα, the transcript of which is downregulated in MTM mice (which is the opposite of the protein levels) but unchanged with tamoxifen. One exception is the progesterone receptor, which is decreased in MTM mice as compared to wild type, and then increased following tamoxifen treatment. Of note, ERα mRNA levels were verified by qRT-PCR (Supplementary Figure 5). We therefore conclude that tamoxifen is not acting via a transcriptional mechanism to improve the MTM phenotype.

**Estrogen receptor alpha localization in *Mtm1* KO mice**. While most investigations concerning ERα center on its role as a transcriptional modulator, there is also recognition that ERα performs extra-nuclear functions[25]. To begin to address this, we examined ERα localization in skeletal muscle (Fig. 6k). In wild-type mice, we found that ERα is localized primarily to the nucleus. In MTM mice, it is instead highly enriched around the sarcolemmal membrane. This membranous expression is reduced, though not completely eliminated, with tamoxifen treatment. In combination with the transcriptome data, these results suggest that extra-nuclear signaling may be involved as a potential mechanism for ERα and tamoxifen in MTM.

**PIK3C2B levels and function are not altered by tamoxifen**. As transcriptome changes did not obviously explain the therapeutic benefit of TAM in MTM mice, we sought to evaluate non-transcriptional targets. Two modifiers have been identified in *Mtm1* KO mice: genetic knockdown/inhibition of either the lipid kinase *Pik3c2b* or the endocytic GTPase *Dnm2* ameliorates the MTM mouse phenotype[8–10,26]. We therefore interrogated the effect of tamoxifen on the level and/or function of these two proteins.

To examine whether tamoxifen modulates PIK3C2B levels and/or function, we examined PIK3C2B protein levels, PI3P levels, PIK3C2B signaling in vitro, and PIK3C2B enzyme activity. As compared to wild type and untreated MTM mice, tamoxifen and estradiol did not alter PIK3C2B protein or mRNA levels (Fig. 7a–h). To evaluate PI3P, one of the lipid products produced by PIK3C2B kinase activity, we measured PI3P levels by immunostaining (Fig. 7i) and found no detectable difference in MTM mouse muscle with tamoxifen exposure. We examined PI3P dependent, MTM1-related membrane trafficking pathways in vitro using HeLa cells (which are documented to express estrogen receptor[27]). Unlike the effect of knockdown of *Pik3c2b*[27], tamoxifen treatment of *Mtm1* siRNA exposed HeLa cells was unable to reverse the defective endosomal exocytosis associated with MTM1 deficiency (Supplementary Figure 6). Finally, we determined whether tamoxifen could directly inhibit

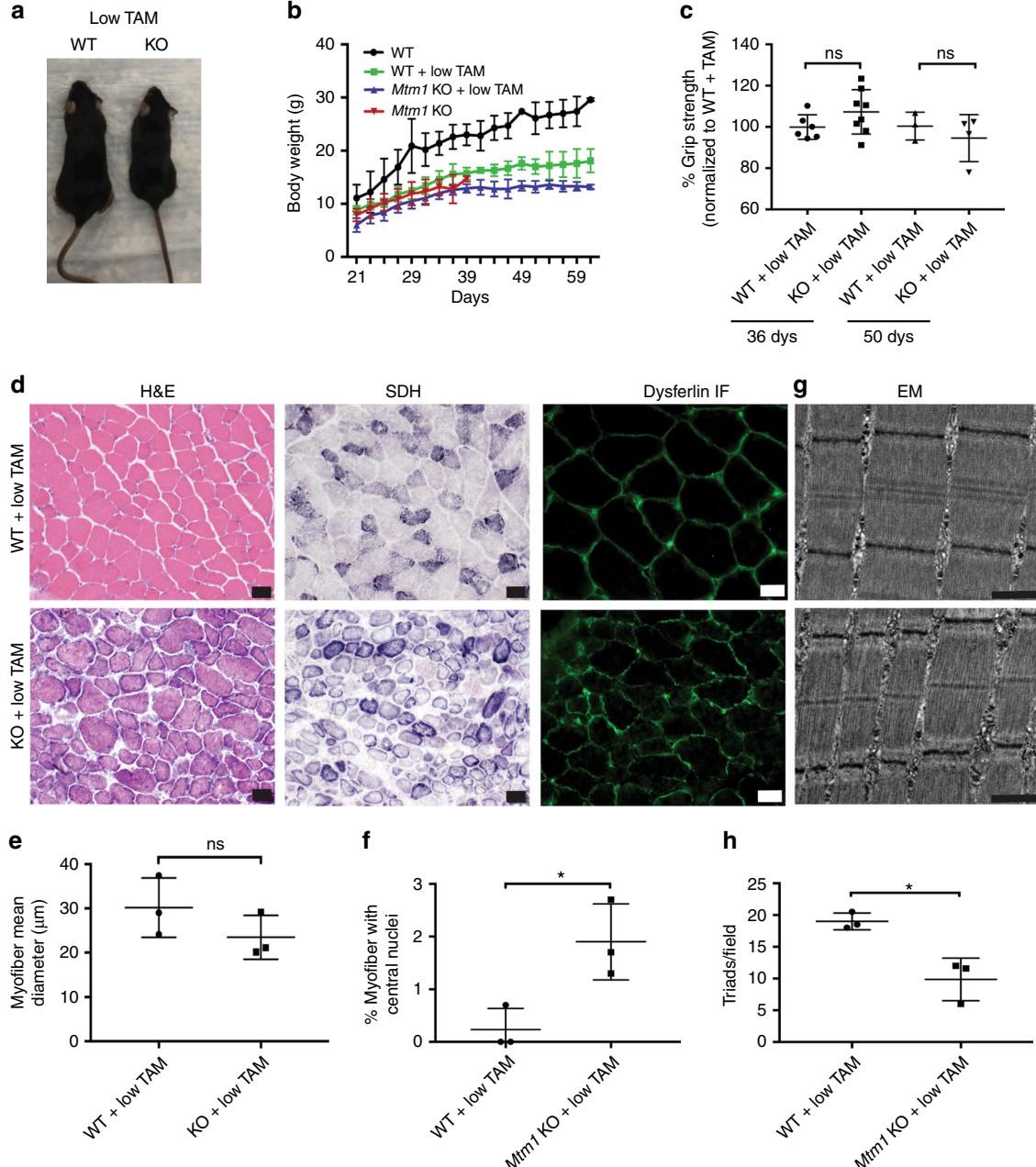

**Fig. 3** Early low-dose tamoxifen treatment prolongs survival and improves muscle strength and structure of *Mtm1* KO mice. **a** Photomicrograph of littermates at 28 days of WT, and *Mtm1* KO treated with low-dose tamoxifen starting 14 days of age. KO treated with low-dose tamoxifen have normal hindlimb positioning and appear similar in size to WT littermates. **b** Plots of body weight show that low-dose treated KOs ($n = 5$) starting at 14 days have similar weight as compared to low-dose treated WT ($n = 3$) and untreated KOs. **c** Early TAM treated KOs have improvement in muscle strength when analyzed at 36 and 50 days of age. At 36 days, TAM treated KOs ($n = 8$) have a mean grip strength of 107 ± 4% vs. TAM treated WT of 100 ± 2% ($n = 6$, *p*-value non-significant). At 50 days, TAM treated KOs ($n = 4$) have a mean grip strength of 95 ± 6% vs. TAM treated WT of 100 ± 4% ($n = 3$, *p*-value non-significant). Data normalized to the mean grip strength of WT + TAM littermates. **d** Early TAM treatment restores *Mtm1* KO muscle structure. Cross-sections from tibialis anterior muscle tissue taken at 36 days and stained for H&E, SDH, and dysferlin (scale bars 20 μm). Early TAM treatment improves overall appearance and myofiber size but not % central nuclei. **e** Myofiber size for KO + TAM = 24 ± 3 μm ($n = 3$) vs. WT + TAM = 30 ± 4 μm ($n = 3$, *p* non-significant). **f** Average percent of central nuclei (per 100 fibers) in KO + low-dose TAM = 2 ± 0.4% vs. WT + low-dose TAM = 0.2 ± 0.1% ($n = 3$, *$p$ = 0.03). **g** Electron microscopy reveals increased triad number in early treated *Mtm1* KOs vs. untreated KOs (scale bars 500 nm). However, triad number is decreases as compared to early treated WT. **h** Quantification of number of triads per field for WT + low-TAM = 19 ± 1.0 ($n = 3$) vs. KO + low TAM = 10 ± 2.0 ($n = 3$, *$p$ < 0.02 compared to WT + TAM). For reference, untreated *Mtm1* KOs have 5 ± 1.5 ($n = 5$) (Fig. 3b)

PIK3C2B kinase activity. This was evaluated using purified PIK3C2B and an in vitro kinase activity assay. At concentrations of tamoxifen equivalent to and exceeding those used in our mice, we did not detect inhibition PIK3C2B kinase activity, with the IC$_{50}$ of tamoxifen against PIK3C2B = 40 μM (compared to IC$_{50}$ = 500 nM of wortmannin, a positive control) (Fig. 7j–k). Together, these results suggest that tamoxifen rescue of MTM mice is not mediated by inhibition of PIK3C2B.

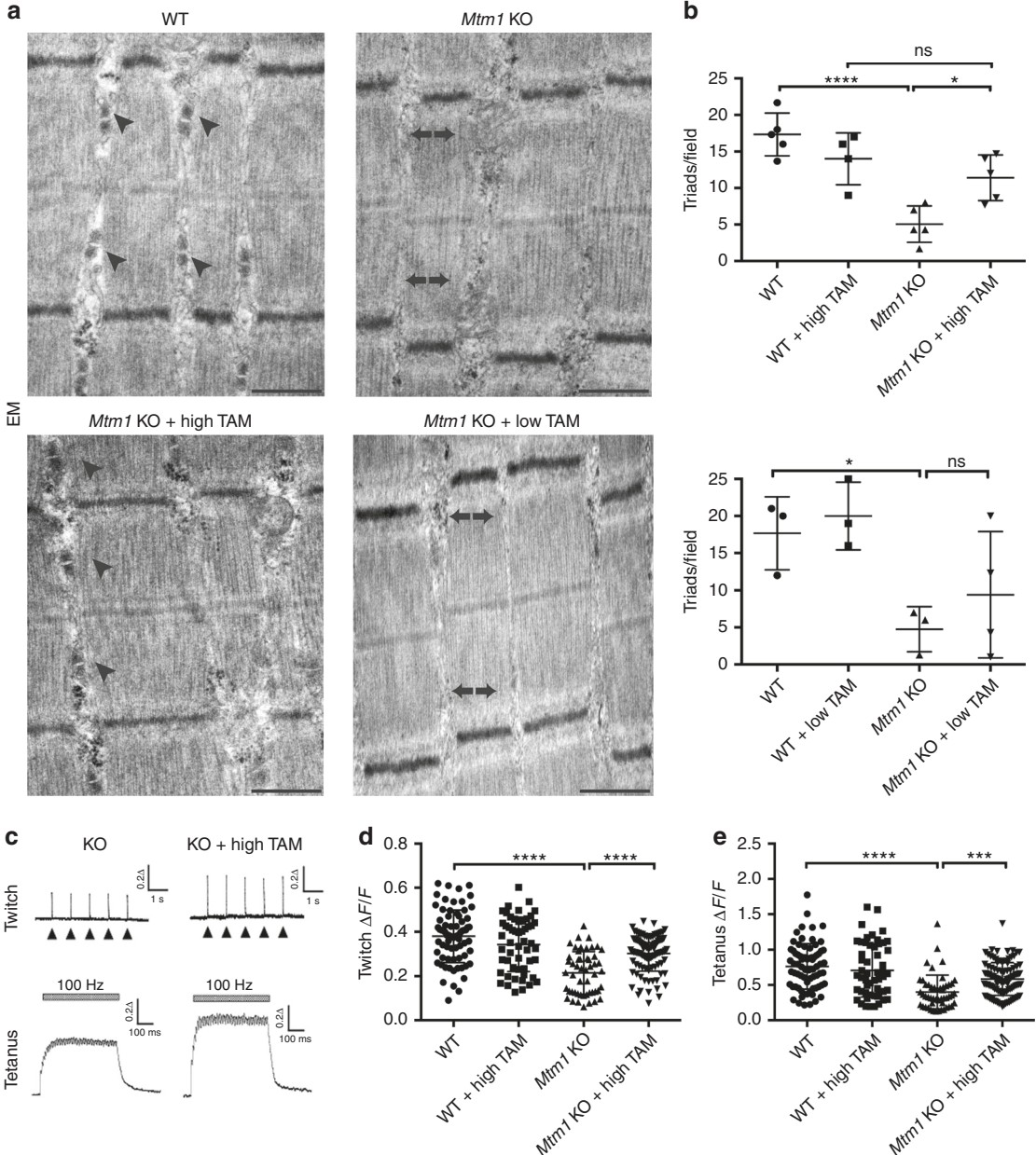

**Fig. 4** Tamoxifen treatment restores triad structure and function in *Mtm1* KO mice. **a** Electron microscopy reveals normal triad structure in WT (arrowheads), reduction of triads in *Mtm1* KOs (arrows), and restoration of triads in high-dose TAM treated KOs (arrowheads). Triad number was not improved in low-dose TAM treated KOs (arrows). **b** Quantification of triad number per field for WT = 17 ± 2 (n = 5), KO = 5 ± 1.5 (n = 5, ****p <0.0001 compared to WT), high-dose TAM treated WT = 14 ± 2 (n = 4), and high-dose TAM treated KOs = 11 ± 2 (n = 5, *p = 0.03 vs. KOs and non-significant vs. WT + TAM). Triad number for low-dose TAM treated KOs = 9.4 ± 5 (n = 4), untreated KOs = 4.8 ± 3 (n = 3, *p < 0.05 compared to WT), WT = 18 ± 5 (n = 3) and WT + low-dose TAM = 20 ± 5 (n = 3). **c** Calcium transients are increased in dissociated muscle fibers from *Mtm1* KO mice treated with high-dose TAM in response to electrically evoked twitch and tetanus stimulation. Representative Mag-Fluo-4 fluorescence traces of FDB fibers during five successive single electrically evoked (Twitch) stimuli and Tetanus (500 ms at 100 Hz) stimulation from either *Mtm1* KO, or KO treated with high-dose TAM. **d** Twitch stimulation is significantly reduced in *Mtm1* KOs (0.21 ± 0.01, n = 51 fibers, ****p < 0.0001 vs. WT = 0.38 ± 0.08, n = 73 fibers) and improved with high-dose TAM (0.30 ± 0.01, n = 102 fibers, ****p < 0.0001 vs. KO). **e** Tetanus stimulation is significantly reduced in KOs compared to WT and improved with high-dose TAM (KO + TAM = 0.60 ± 0.03, n = 103 fibers, ***p < 0.001 vs. untreated KO = 0.40 ± 0.05, n = 51 fibers, ****p < 0.0001 vs. WT = 0.8 ± 0.04, n = 73 fibers, and vs. WT + TAM = 0.70 ± 0.05, n = 53 fibers). Statistical analyses were conducted by two-way ANOVA, followed by Tukey's multiple comparisons post-test or Fisher's least common differences post-test. For direct two sample comparisons, unpaired, parametric two-tailed Student's *t*-test was performed. Scale bars = 500 nm

**Tamoxifen reduces DNM2 levels in vivo and in vitro**. DNM2 protein levels (but not transcript levels) are elevated in *Mtm1* KO mice and lowering these levels ameliorates most aspects of the MTM mouse phenotype[9,26,28]. Thus, we quantified DNM2 levels in the context of tamoxifen treatment. Importantly, tamoxifen therapy at either low or high-dose significantly reduced DNM2 protein levels in MTM skeletal muscle (Fig. 8a–f), and high-dose therapy reduced levels in wild-type mice as well (Fig. 8a–e). Estradiol, but not fulvestrant, also reduced DNM2 protein levels in MTM mice, though not to

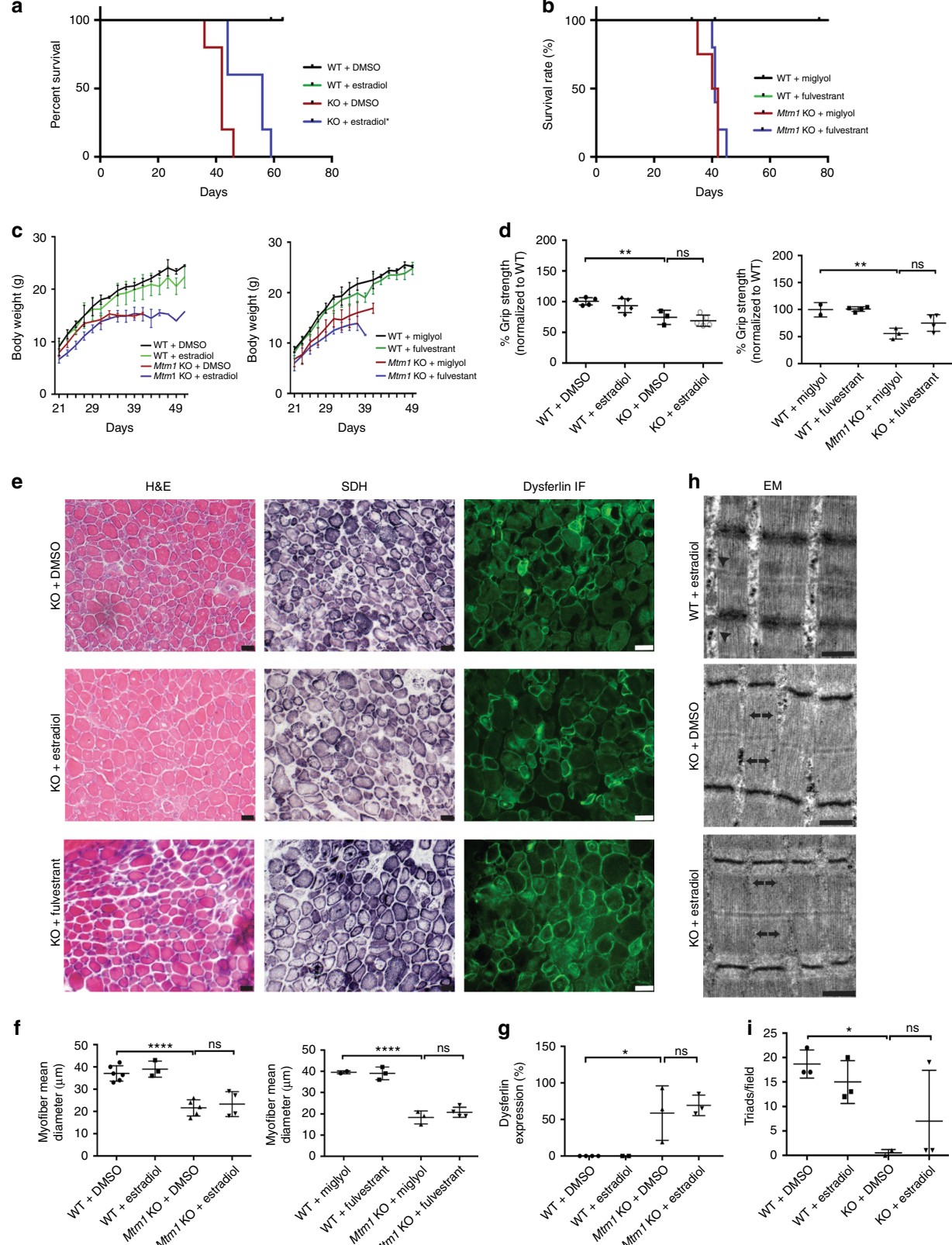

levels of wild-type littermates (Fig. 8c–h). Of note, tamoxifen treatment did not change *Dnm2* transcript levels (Supplementary Figure 7), suggesting the effect of tamoxifen on DNM2 is mediated via changes in DNM2 protein translation and/or stability.

To verify that the effect on DNM2 by tamoxifen is not secondary to other improvements of muscle structure and function in MTM mice, and to assess the relevance of our data to the human disease, we investigated tamoxifen effect in fibroblasts and transdifferentiated myotubes derived from MTM

**Fig. 5** Estradiol, but not fulvestrant, improves survival in *Mtm1* KO mice. **a** 17β-estradiol treatment starting at 21 days (1 μg/day) improves survival from 42 days for KOs to 56 days ($n = 5$, *$p < 0.05$), with longest survival to 59 days. **b** Fulvestrant treatment starting at 21 days (1 mg/kg, $n = 5$) does not improve survival of *Mtm1* KOs, with median survival of 41 days for both KO + fulvestrant ($n = 5$) and KO ($n = 4$) alone. **c** *Mtm1* KO body weight is not restored with either estradiol or fulvestrant. **d** 17β-estradiol treatment of *Mtm1* KOs does not improve muscle strength. Estradiol treated KOs ($n = 5$) mean grip strength is reduced to 69 ± 6%, untreated KOs to 74 ± 7% ($n = 3$) (**$p = 0.0020$ compared to WT), while estradiol treated WT are 93 ± 7% ($n = 5$). Values are normalized to WT + DMSO (100 ± 6%, $n = 5$). Fulvestrant treatment of KOs also does not improve muscle strength. Fulvestrant treated KOs ($n = 4$) mean grip strength = 75 ± 9%, untreated KOs = 56 ± 8% ($n = 3$) (**$p = 0.0021$ compared to WT), fulvestrant treated WT = 101 ± 8% ($n = 4$). Values normalized to WT + miglyol (100 ± 7%, $n = 2$). **e** Estradiol and fulvestrant treatment do not improve *Mtm1* KO muscle structure. As with untreated KOs, muscle from KOs treated with either estradiol or fulvestrant show decreased myofiber size, increased mitochondrial aggregation, and abnormal distribution of dysferlin staining. **f–i** Quantification of muscle structural parameters. There was no significant change in *Mtm1* KO myofiber size with estradiol or fulvestrant treatment **f**, and no significant change in dysferlin distribution **g** or triad number with estradiol treatment. The full enumeration of all data points is presented in Supplementary Table 3. Statistical analyses were conducted by two-way ANOVA, followed by Tukey's multiple comparisons post-test or Fisher's least common differences post-test. For direct two sample comparison, unpaired, parametric two-tailed Student's *t*-test was performed. Scale bars = 20 μm

patients (Fig. 8i–l and Supplementary Figures 8, 9). Similar to MTM mice, untreated patient fibroblasts and transdifferentiated myotubes exhibit increased levels of DNM2 (Fig. 8i–j). Exposure of these cells to 10 μm tamoxifen produced a significant decrease in DNM2 (Fig. 8k–l), with dose dependent reduction observed in patient fibroblasts at 15 and 20 μm (Supplementary Figure 10).

**DNM2 reduction by TAM is *Mtm1* KO independent and ER dependent**. To test whether the effect of tamoxifen on DNM2 requires estrogen receptor and/or was present only in the setting of *Mtm1* deficiency, we examined several additional cell lines. C2C12 myotubes showed a dose dependent reduction in DNM2 protein levels with tamoxifen (Fig. 8m–o and Supplementary Figure 11). This myoblast line primarily expresses an extra-nuclear ERα (Supplementary Figure 12A). MCF7 cells also express ERα, and tamoxifen is a well-established modulator of ER signaling in these cells[29]. When treated with tamoxifen, MCF7 cells also had a dose dependent lowering of DNM2 levels (Fig. 8q–s). In contrast, HEK293T cells, which lack ERα (Supplementary Figure 12B), did not demonstrate changes in DNM2 levels when treated with tamoxifen (Fig. 8n–p). Taken together, these in vitro data support the conclusion that tamoxifen reduces DNM2 protein levels in multiple cell lines in an ERα-dependent manner.

**DNM2 reduction by tamoxifen is prevented by blocking the UPS**. ERα-mediated activation of the ubiquitin-proteasome system (UPS) represents an important feedback mechanism to reduce ERα levels, and estradiol and tamoxifen modulate the UPS via both transcriptionally mediated and extra-nuclear mechanisms[30]. Correspondingly, and as noted above, we found ERα protein levels reduced following both high and low-dose tamoxifen treatment, as well as by estradiol (Fig. 6). Given that transcript levels are not changed, this result is consistent with tamoxifen-induced UPS activation[31]. Therefore, we hypothesized that tamoxifen modulates DNM2 levels via UPS activation. To test this idea, we treated MCF7 cells with the UPS inhibitors MG-132 and bortezomib (Fig. 8r–t). In the absence of tamoxifen, both drugs increased DNM2 protein levels. With tamoxifen exposure, both drugs prevented the TAM associated decrease in DNM2, with resulting levels similar to untreated cells. These data support an interplay between the UPS and TAM-dependent DNM2 regulation.

## Discussion

In this study, we describe the identification of the FDA approved drug tamoxifen as an effective therapy in a mouse model of X-linked myotubular myopathy (MTM). Our results uncover the first small molecule therapeutic with pre-clinical efficacy and potential clinical translatability for myotubular myopathy. Given the widespread experience with tamoxifen in pediatric patients, including settings such as central nervous system tumors and precocious puberty[32,33], our data provide the necessary pre-clinical evidence to translate these findings into a clinical trial of tamoxifen in MTM patients.

The most important implications of our findings relate to the potential of tamoxifen as a treatment for MTM, an orphan disease with devastating clinical consequences that is currently without therapy. While other strategies are under development, tamoxifen represents the only FDA approved medication in consideration. As such, it presents the opportunity for rapid development, as no additional toxicology and compound refinement are necessary, as would be the case for other strategies such as antisense oligonucleotide based DNM2 knockdown or PIK3C2B inhibitor therapy[8,9]. AAV8 mediated *MTM1* gene replacement therapy[7] has just entered clinical trial, and may provide substantial clinical improvement for MTM patients. However, this is currently targeted at individuals less than 4 years of age, and its benefit is unknown. Given that tamoxifen is safe, well tolerated and inexpensive, if it proves effective in MTM patients, it may serve either as a stand-alone therapy (for those not eligible for gene therapy) or as a valuable adjunctive treatment. Other strengths of tamoxifen as a potential MTM therapeutic include that it modulates a known disease modifier (DNM2) in a dose dependent manner, and that it can do this in both a mouse model and in patient derived fibroblasts and myotubes.

MTM is one subtype of a broader group of muscle conditions called centronuclear myopathies (CNM). There are considerable pathomechanistic overlaps between centronuclear myopathies, particularly in relation to triad defects and alterations in DNM2 protein levels. In fact, genetic reduction of DNM2 levels has been demonstrated to improve phenotypic and pathologic features in mouse models of MTM and *BIN1* related CNM[34]. It is thus tempting to speculate that tamoxifen may also be effective for this broader class of muscle diseases. Future experiments will be necessary to address this possibility.

As our discovery of tamoxifen as an MTM disease modifier was serendipitous, we sought to define how the drug worked to ameliorate disease progression. Our data illuminate several points in this regard. We show that tamoxifen is working primarily via estrogen receptor pathway(s), as supported by the ability of estradiol (but not fulvestrant) to improve the mouse phenotype and by our data from cell lines of diverse tissue origin. We further demonstrate that this action is likely mediated by extra-nuclear estrogen receptor alpha signaling, as revealed by our expression and localization studies and our RNA-seq analysis. We also show

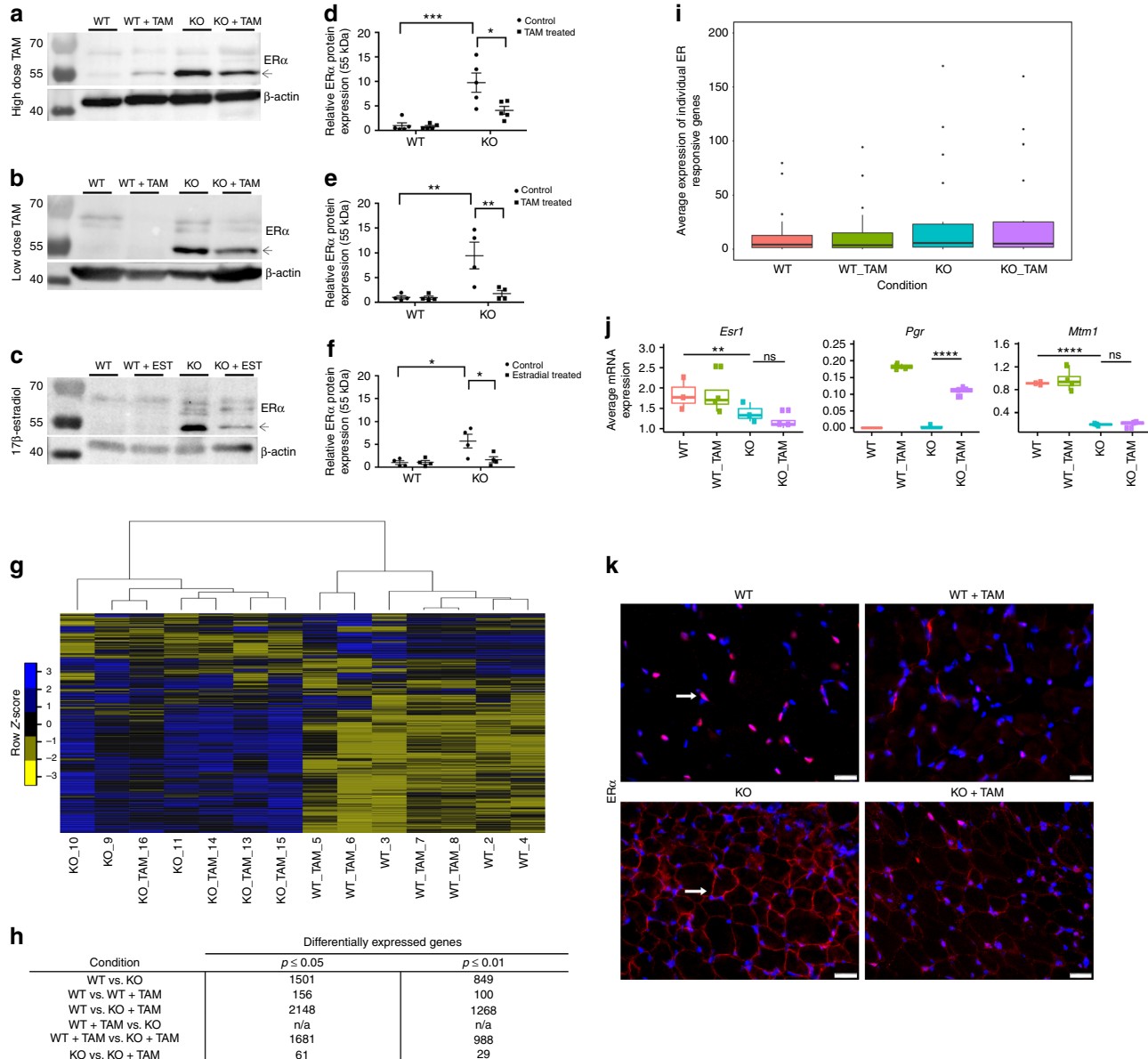

**Fig. 6** Tamoxifen treatment does not alter the *Mtm1* knockout transcriptome. **a–c** Representative western blots for ERα (with β-actin loading control) for WT, WT + TAM, *Mtm1* KO, and tamoxifen (TAM) treated *Mtm1* KO mice (molecular weight markers indicated in kDa). **d–f** Quantification of protein levels of the 55 kDa isoform of ERα as determined by densitometry with standardization to β-actin and represented as fold difference from the average of WT ($n$ = 5 mice per group for high-dose TAM, $n$ = 4 mice per group for low-dose TAM and estradiol. Statistical analyses were conducted by two-way ANOVA, followed by Tukey's multiple comparisons post-test *$p < 0.05$, **$p < 0.01$, ***$p < 0.001$, ****$p < 0.0001$. Complete data points presented in Supplementary Table 4. **g** Heat map indicating overall mRNA expression profiles (derived from RNA-seq) of the four experimental groups, blue = high expression, yellow = low expression. *Mtm1* KO and *Mtm1* KO + TAM heat maps cluster together and are distinct from WT and WT + TAM. **h** Table indicating the total number of gene transcripts in each comparison with significantly differentially expression ($n$ = 3 mice per group for WT, $n$ = 4 for WT + TAM, $n$ = 3 for KO, and $n$ = 4 for KO + TAM). Note that only 29 genes are differentially expressed ($p \leq 0.01$, DEseq and EdgeR R/Bioconductor pairwise comparisons) in *Mtm1* KO vs. *Mtm1* KO + TAM. **i** Box plot indicating the average expression of 21 estrogen receptor-responsive genes within each experimental group; no overall differences are observed with TAM treatment. **j** Individual expression box plots of *Esr1* and *Pgr1*, the only significantly changed estrogen response genes, and of *Mtm1*. Box plots show the median, the 25th and 75th percentiles, and outliers. **k** Immunofluorescence with anti-ERα (red; indicated by a white arrow) on tibialis anterior muscle sections from WT, WT + TAM, *Mtm1* KO, and *Mtm1* KO + TAM (blue = DAPI) ($n$ = 3 mice per group for WT and WT + TAM, $n$ = 4 mice per group for KO and KO + TAM). ERα is expressed primarily in the nucleus in WT muscle (arrow) but found at the sarcolemmal membrane in *Mtm1* KOs (arrow). Overall ERα expression and its membrane localization are reduced with TAM treatment. Scale bars = 20 μm

for the first time that tamoxifen can modulate levels of DNM2 protein, a key MTM disease modifier. We assert that the primary means via which tamoxifen improves the MTM phenotype is by lowering DNM2, a hypothesis supported by our data and by existing studies that use both genetic and antisense oligonucleotide based strategies to lower DNM2 levels in MTM mice[9,20].

Importantly, we observed that tamoxifen's ability to lower DNM2 levels could be prevented by inhibition of the ubiquitin-proteasome system, indicating that the drug may act in MTM mice by increasing proteolysis. This would be in keeping with a known role of tamoxifen (i.e. stimulating proteolytic degradation of ERα)[35], and would be parsimonious with the lack of

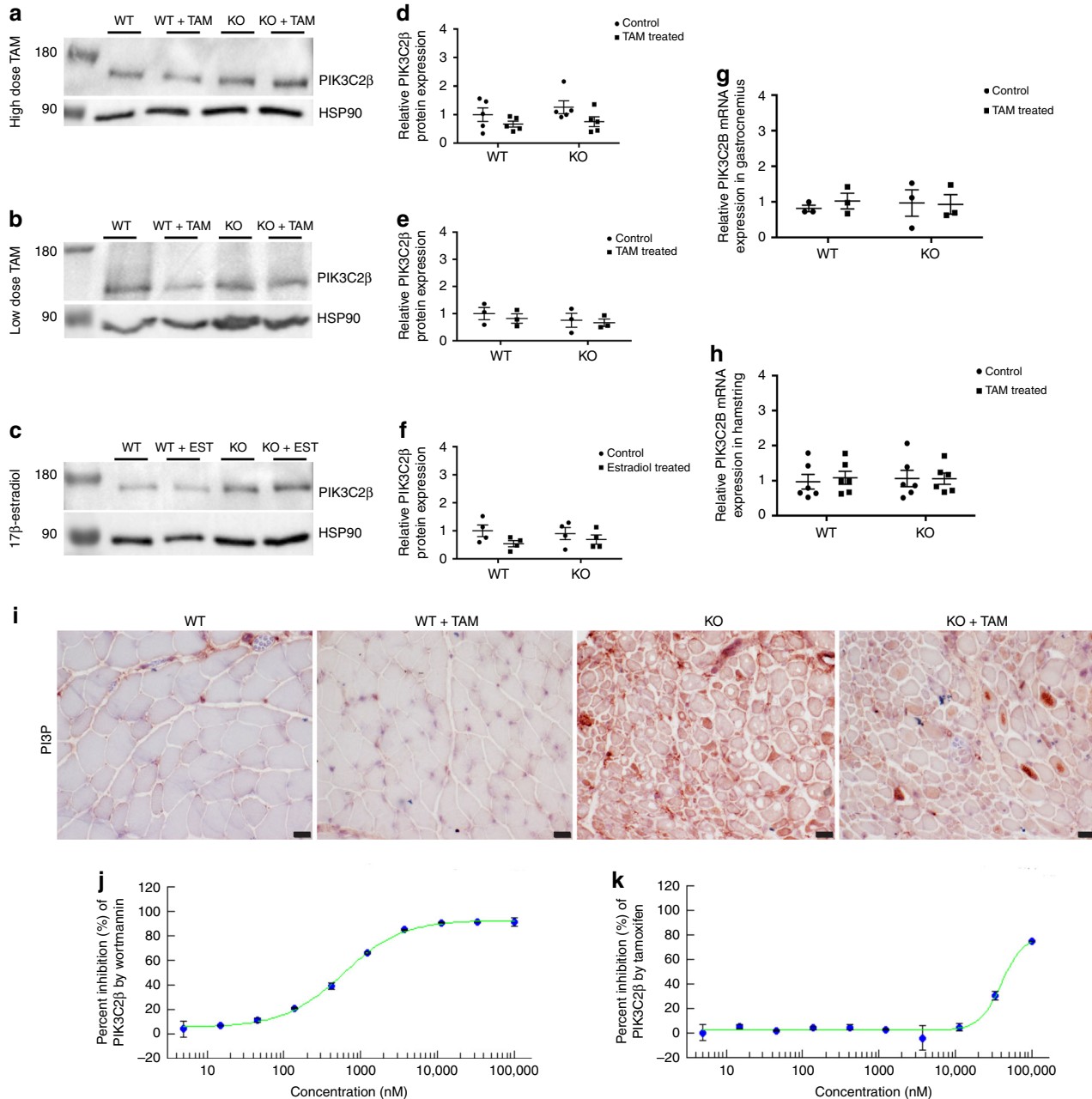

**Fig. 7** Tamoxifen treatment does not affect the levels or activity of PIK3C2B. **a** Representative western blot for PIK3C2B, with HSP90 loading control, in *Mtm1* KO mice treated with high-dose tamoxifen (TAM), **b** low-dose TAM, and **c** 17β-estradiol; position of the molecular weight markers indicated (in kDa). Samples for each drug trial derive from the same experiment, and blots were processed in parallel for each drug trial. **d–f** PIK3C2B levels were determined by densitometry, standardized to HSP90 and are represented as the fold difference from the average of the WT ($n = 5$ mice per group for high-dose TAM, $n = 3$ for low-dose TAM, $n = 4$ for 17β-estradiol); graphs represent, 5, 3, and 4 technical replicates respectively and mean ± SEM). **g, h** Graphs depicting mRNA levels of PIK3C2B in **g** gastrocnemius and **h** hamstring of WT and *Mtm1* KO mice ± tamoxifen (TAM) treatment. mRNA expression was determined by real-time qPCR and analyzed by the $2^{-\Delta\Delta CT}$ method. Values were normalized to actin and represent the fold difference from the average of the WT ($n = 3$ mice per group for gastrocnemius; $n = 6$ mice per group for hamstring, graphs represent three technical and biological replicates for gastrocnemius; three technical and biological replicates for hamstring and ±SEM). Statistical analyses for **a–h** were conducted by two-way ANOVA followed by Tukey's multiple comparisons post-test. No significant differences were observed. **i** Immunohistochemistry against PI3P in WT and *Mtm1* KO mice, and TAM treated WT and *Mtm1* KO mice show no overall difference in PI3P levels following treatment with high-dose TAM. **j, k** Graphs of PIK3C2B in vitro kinase assay depicting the percent inhibition of PIK3C2B by wortmannin (control; $IC_{50} = 500$ nM) and tamoxifen ($IC_{50} = 40$ μM). Scale bars = 20 μm

transcriptional changes seen in MTM mice with tamoxifen therapy. Whether this effect is specific for DNM2 or is more general will require future investigation. Interestingly, DNM2 has been previously shown to be required for ERα transit to the lysosome[36], so it is reasonable to speculate that DNM2 and ERα get degraded together.

Of note, estrogen signaling and tamoxifen have been previously identified as modulators of skeletal muscle structure and function. Reduction of estradiol levels in females is associated with reduced strength and muscle wasting[37], and muscle specific knockout in mice of estrogen receptor alpha results in impaired force generation[38]. Conversely, hormone replacement therapy has been

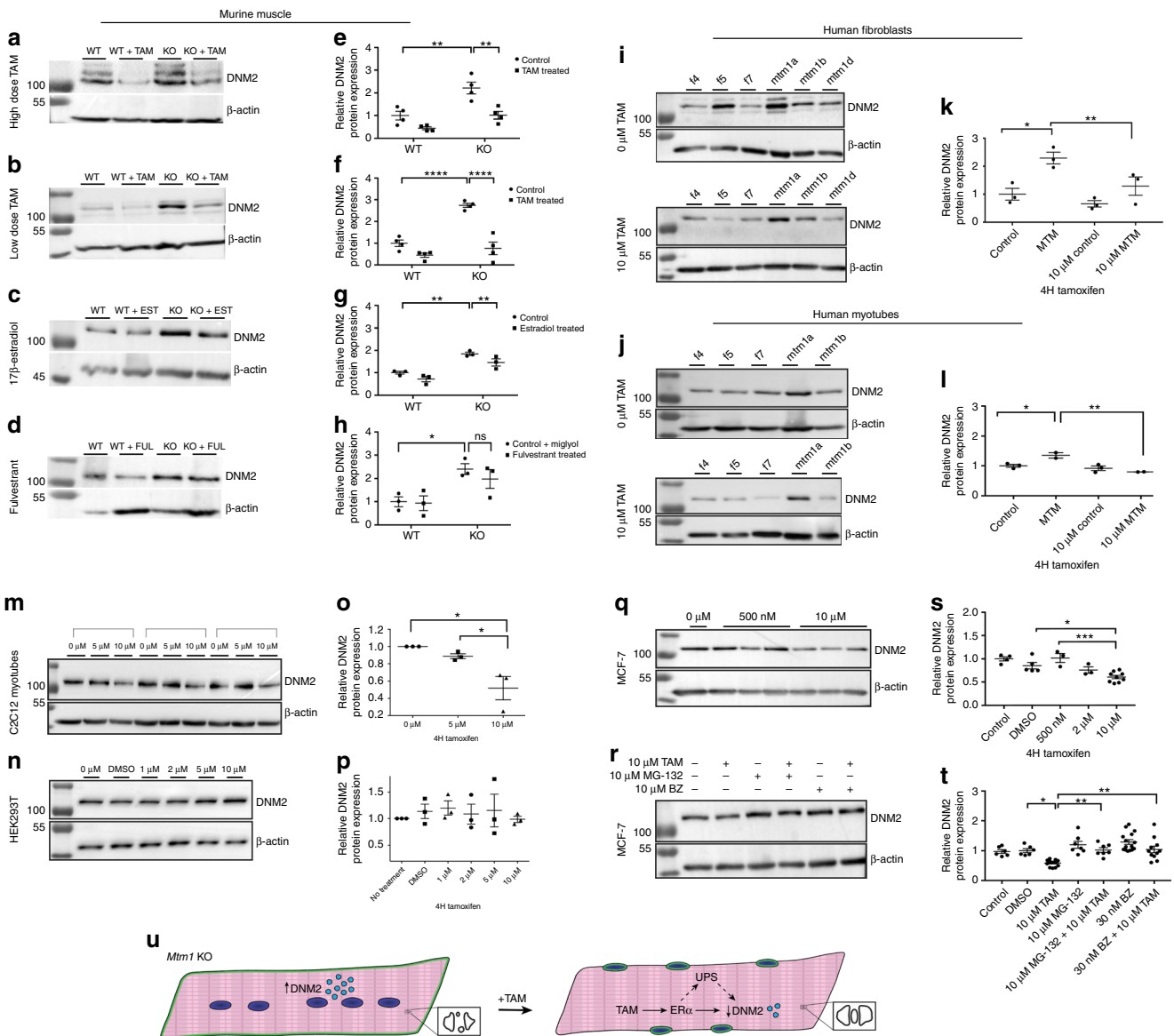

**Fig. 8** Tamoxifen reduces dynamin-2 protein level in vivo and in vitro. **a–d** Representative western blots for dynamin-2 (DNM2) (β−actin loading control) for *Mtm1* KO mice ± tamoxifen (TAM). **a, b** 17β-estradiol **c** and fulvestrant **d**. **e–h** DNM2 protein levels are increased in *Mtm1* KO and reduced by TAM and estradiol treatments. Quantification determined by densitometry (standardized to β-actin) and represented as fold difference from WT (*n* = 4 mice per group for high and low-dose TAM; *n* = 3 for 17β-estradiol and fulvestrant); graphs represent 4 and 3 technical replicates respectively, and mean ± SEM. **i–q** Representative western blots for DNM2 in various cell culture models, with accompanying quantification (presented as fold difference vs. average for untreated cells). **i–l** DNM2 is increased in human MTM patient fibroblasts (*n* = 3 cell types per group for control and MTM) and transdifferentiated myotubes (*n* = 3 cell types per group for control and *n* = 2 for MTM) and decreased with TAM; graphs represent three technical replicates and mean ± SEM. **k, l** DNM2 can be reduced with TAM in C2C12 myotubes **m, o** and MCF-7 cells **q, s** both of which express ERα (Supplementary Figure 12A), but not in HEK293T cells **n, p** which do not (Supplementary Figure 12B); graphs represent three technical replicates and mean ± SEM. **r** Representative western blot for DNM2 for MCF-7 cells treated with 10 μM MG-132 and 30 nM Bortezomib (BZ) ±10 μM TAM; graphs represent 6–15 technical replicates and mean ± SEM. **t** Both UPS inhibitors block TAM's effect on DNM2, as revealed by densitometric quantification (values standardized to β-actin and represented as the fold difference from the average of untreated control). Statistical analyses were conducted by two-way ANOVA for mouse experiments, and one-way ANOVA for in vitro experiments, followed by Tukey's multiple comparisons post test. *$p \leq 0.05$, **$p \leq 0.01$, ***$p \leq 0.001$, ****$p \leq 0.0001$. Of note, a full listing of all data points for **a–r** is presented in Supplementary Table 5. **u** Schematic representing proposed mechanism of action of TAM in *Mtm1* KO muscle

shown to increase strength in post-menopausal women[39], and pre-clinical studies have demonstrated a role for estrogen signaling in protecting against oxidative stress[40,41], in maintenance of mitochondrial function[42], and in post-translational modulation of the contractile apparatus[38]. In addition, both estradiol and tamoxifen can modulate ER-dependent gene expression and activate downstream signaling pathways in skeletal muscle[43,44].

Of most relevance to our study, Dorchies et al. have previously shown that tamoxifen can improve pathology and increase force generation of the mdx mouse model of Duchenne muscular dystrophy (DMD)[45]; Wu et al. have similarly demonstrated improvements with tamoxifen in a mouse model of *FKRP*-related muscular dystrophy[46]. The mechanism(s) through which this effect on muscular dystrophy is exerted are not clear, in keeping

with the fact that mode(s) of action by which tamoxifen acts, particularly in male skeletal muscle, are incompletely understood.

Interestingly, in our study, tamoxifen, at least at high dose, appears to lower DNM2 protein levels in wild-type mouse muscle, and additionally in several cell lines with normal MTM1 protein levels, including the C2C12 myotube line. We correspondingly observed a small but significant reduction in myofiber size in wild-type male animals treated with high-dose tamoxifen (though no change in EC coupling or grip strength), suggesting that tamoxifen therapy may have a direct consequence on normal muscle structure. The potential clinical impact of this observation is not certain, and tamoxifen in male patients is generally very well tolerated. In a recent meta review of clinical trials involving nearly 1000 male patients treated with tamoxifen, 5% of patients reported excessive fatigue, 2% complained of myalgias, and 3% reported gait abnormalities[47]. Any potential interplay between dose dependent tamoxifen response and skeletal muscle-related phenotypes in patients treated with the drug likely merits additional systematic examination.

We also noted, in a fraction of MTM mice treated with high-dose tamoxifen starting at a very late stage of the disease (30 days), precipitant death due to pulmonary hemorrhage. There are also a small number of case reports of acute and severe lung injury in adult patients treated with tamoxifen[48]. The mechanism underlying this effect, either in our mice or in cancer patients, is not clear. However, given that we observe this effect in only a small number of late stage, high dose treated MTM mice, and that our "low-dose" treatment effectively promotes survival and improves motor function and more closely mirrors typical dosing in humans, we do not anticipate this impacting the translatability of the drug for MTM patients. That stated, it will likely be important to monitor for pulmonary injury in any clinical studies of tamoxifen for MTM.

Finally, while our data support a clear estrogen receptor mediated effect of tamoxifen in MTM, it is possible that tamoxifen has non-ER-related effects as well. The fact that only high-dose tamoxifen improves triad structure and function in MTM mice is consistent with this assertion. Future experimentation is necessary to investigate this intriguing observation. It is also of interest that low-dose therapy, despite lowering DNM2 levels, improving grip strength, and prolonging survival, did not significantly improve triad structure/function (though there was a trend toward increase in triad number) and did not ameliorate the abnormal histopathologic finding of central nucleation. This suggests that the abnormalities in muscle structure and function in MTM may be driven through multiple aberrant pathways, and opens the door for development of combinatorial therapeutic approaches that address different aspects of the disease.

Importantly, Dorchies and colleagues have simultaneously and independently demonstrated efficacy of tamoxifen in the mouse model of MTM[49]. Their study, performed on mice with the same mutation in *Mtm1* but maintained on a different genetic background (129/agouti vs. C57B6J in our study) corroborates our data, showing significant improvements in muscle structure and function and dramatic prolongation of survival. Taken together with our results, their data further support the potential translatability of tamoxifen for MTM.

In conclusion, we identify tamoxifen as a potent, FDA approved small molecule modifier of the MTM mouse phenotype. While additional work will continue to unravel the mechanistic interplay between tamoxifen, ER signaling, and the MTM disease process, our findings have immediate clinical implications. Chronic tamoxifen treatment is safe and well tolerated in children, and thus translation to MTM patients (facilitated by our ongoing work on MTM natural history) through a clinical trial is warranted and our proposed next step.

## Methods

**Animal care and treatment**. All animal procedures were performed in compliance with the Animals for Research Act of Ontario and the Guidelines of the Canadian Council on Animal Care. The Centre for Phenogenomics (TCP) Animal Care Committee reviewed and approved all procedures conducted on animals at TCP. Mice were maintained in specific pathogen free conditions (SPF) under proper environmental regulations: temperature and light cycles, unlimited access to water, appropriate food supply, and clean enclosures. Pups were weaned from their mothers according to standard protocols, and tails were clipped for genotyping.

**Generation and genotyping of MTM mouse strains**. *Mtm1* knockout mice with targeted deletion of exon 4 of the *Mtm1* gene were a generous gift from Dr. Anna Buj-Bello. Genotyping protocols and primers for *Mtm1* mice were conducted and used as previously described[11]. *Mtm1* KO mice are maintained on a C57BL6J background and backcrossed approximately every 2–3 generations.

**Drug treatments**. Drug treatments in this study were conducted starting from 14, 21, or 30 days of age and continued until 36 days of age or until animals reached endpoint (defined as lack of movement, significant weight loss, or hindlimb paralysis). For 21 and 30 day starting points, animals were randomized for treatment vs. placebo after genotypes were confirmed (just prior to 21 days). We estimated the number of animals needed by performing a theoretical power calculation based on untreated KO survival (mean values at the time of calculation 37 ± 5). We determined a minimum number needed per group of 8 to detect a 10-day improvement in survival with 95% confidence and $p < 0.05$. Tamoxifen-citrate mixture at 40 mg/kg (high-dose) or 3 mg/kg (low-dose) was incorporated into the standard rodent diet premixed with ~5% sucrose as a palatability enhancer (Harlan Laboratories Teklad Diets). 17-β Estradiol (E2, Sigma-Aldrich) was given at 1.0 μg/mouse/day dissolved in 0.1% DMSO administered by subcutaneous injections, once daily excluding weekends. Fulvestrant (ICI 182 780, Sigma-Aldrich) was given at 1 mg/kg dissolved in Miglyol 812 (OmyaPeralta GmbH) administered by subcutaneous injection, once daily excluding weekends. Age-matched litter control WT and *Mtm1* KO males were injected with a corresponding solution of 0.1% DMSO or Miglyol. Drug and placebo administration was performed in a genotype-blind fashion based on the animal ID number. Body weight and food consumption were monitored three times weekly. For tamoxifen treatment starting at 14 days of age, low-dose tamoxifen containing food was given to lactating moms until weaning age (21 days). After 21 days, weaned MTM KOs and littermate controls were then kept on the low-dose diet as described above for the 21-day treatment group. Animals were kept on tamoxifen until 36 days (for histopathological analyses) or until humane endpoint was reached.

**Phenotypic studies**. Grip strength analyses were performed using the Bioseb-Bio-GS3 at the Clinical Phenogenomics Core facility at the Toronto Centre for Phenogenomics as per standard protocol. Examiners were blinded to genotype and treatment status.

**Histology and staining analysis**. Muscle tissue from tibialis anterior (TA) was dissected and mounted into small balsawood pieces previously frozen with drops of optimal cutting temperature compound (Tissue-Tek; Thermo Fisher). Mounted muscle tissues were then flash frozen in isopentane at −55 °C. Eight micrometres of cross-sections of flash frozen TA muscle was cut and mounted on Superfrost Plus slides (Thermo Fisher) using a Leica cryostat (Leica Microsystems) at −20 °C. For indirect immunofluorescence, mounted TA sections were fixed in 4% paraformaldehyde for 15 min at room temperature, and blocked for 30 min at room temperature in blocking buffer (1% Tris-buffered saline (TBS) solution, 0.1% bovine serum albumin, 10% goat serum, 0.1% Triton X-100, and 0.1% Tween-20 (pH 7.9). Slides were then incubated overnight at 4 °C with primary antibodies against Dysferlin (1:1000; Abcam), or estrogen receptor alpha (ERα, 1:100; SC-7207, Santa Cruz) in blocking buffer. For fiber typing, double immunofluorescence was performed using antibodies against Dystrophin (1:500; ab15277, Abcam), myosin heavy chain type 1 (Skeletal, Slow, 1:50; Sigma-Aldrich), fiber type 2a (1:50; clone SC-71) or fiber-type 2b (1:50; clone BF-F3, Developmental Studies Hybridoma Bank) overnight at room temperature. Secondary antibodies (anti-mouse Alexa Fluor® 488 1:1000; or anti-rabbit Alexa Flour® 555 1:1000, Life Technologies) were applied for 1 h at room temperature the following day. Slides were washed several times to remove excess secondary antibody and then mounted with ProLong Gold (Life Technologies) or VECTASHIELD with 4′,6-diamidino-2-phenylindole (DAPI) (Vectorlabs). Micrographs were captured with an Infinity1 camera (Lumenera Corporation) with eponymous software visualized through an Olympus BX43 light microscope (Olympus). All measurements and quantification were performed by staff blinded to genotype and treatment status. Slides were labeled with coded numbers only to prevent interruption of the blind.

**Bright-field imaging and fiber size analysis**. Slides were stained with Mayer's H&E or SDH following standard protocols, and then mounted with Permount (Thermo Fisher). Quantification of myofiber size and number of centrally nucleated fibers was performed manually. Fiber-type quantification was performed using

an automated method (Imaris Image Analysis software). Blinding was as described in the previous section.

**Transmission electron microscopy (TEM)**. TA muscle samples for TEM were fixed in 2% glutaraldehyde containing 0.1 M sodium cacodylate buffer overnight at 4 °C then taken to the Advanced Bioimaging Center (The Hospital for Sick Childen). After rinsing in buffer, samples were post-fixed in 1% osmium tetroxide, dehydrated in a graded ethanol series followed by propylene oxide, and embedded in Quetol-Spurr resin. Ninety nanometers of thick sections was cut using an RMC MT6000 ultramicrotome, stained with uranyl acetate and lead citrate, and viewed in an FEI Tecnai 20 TEM.

**Isolation of flexor digitorum brevis (FDB) muscle fibers**. FDB muscle fibers were dissociated from the footpad of 5–6-week-old-male mice by enzymatic digestion with collagenase A (1 mg/ml) in regular rodent Ringer solution (in mM: 146 NaCl, 5 KCl, 2 MgCl$_2$, 2 CaCl$_2$, and 10 HEPES, pH 7.4) with gentle agitation for 1 h at 37 °C. Single FDB muscle fibers were plated on 35 mm glass-bottomed dishes (MatTek Corp) and allowed to settle for 20 min before experimentation. Of note, mice for these experiments were treated in Toronto (Dowling), then shipped to Rochester (Dirksen). Upon receipt in Rochester, animals were identifiable by code only, with the Dirksen lab blinded to genotype and treatment status and with the code maintained until study completion at the Dowling lab.

**Electrically evoked Ca$^{2+}$ release in single FDB fibers**. FDB fibers were loaded with 4 μM Mag-fluo-4 for 20 min at room temperature followed by washout in regular rodent Ringer supplemented with 25 μM benzyl-p-toluene sulfonamide (BTS) for 20 min. Loaded fibers were electrically stimulated with a series of electrically evoked twitch stimulations (1 Hz) or 5 consecutive tetani (500 ms at 100 Hz) using an extracellular electrode placed next to the cell of interest. Mag-fluo-4 was excited at 480 ± 15 nm using an Excite epifluorescence illumination system (Nikon Instruments) and fluorescence emission at 535 ± 30 nm was monitored with a ×40 oil objective and a photomultiplier detection system (Photon Technologies Incorporated, Birmingham, NJ). Relative changes in mag-fluo-4 fluorescence from baseline (F/F0) were recorded using Clampex 9.0 (Molecular Devices).

**Tissue extraction**. Total protein lysates were extracted from skeletal muscle of KO and TAM treated mice, in addition to their age-matched littermate controls at 36 days of age. Quadriceps muscle was snap-frozen in liquid nitrogen and stored at −80 °C until processing. The muscle was minced and homogenized for 3 min using the TissueLyserII (Qiagen) in 1X cell lysis buffer supplemented with protease and phosphatase inhibitors. Lysates were chilled at 4 °C for 10 min, then centrifuged at 12,000×g for 10 min at 4 °C. Supernatants were collected, and protein concentration determined using the Pierce™ BCA protein assay kit (Thermo Fisher Scientific). Extracts were diluted with 4X reducing Laemmli buffer before immunoblotting.

**Immunoblotting**. Muscle protein lysates (60 μg/lane) were resolved by SDS-PAGE, and proteins were transferred onto nitrocellulose membranes, according to standard procedures. Equal loading and transfer efficiency were verified by Ponceau S Red (Sigma) staining. Membranes were blocked for 1 h in 1X TBST (20 mmol/L Tris-base, 150 mmol/L NaCl, 0.1% Tween-20, pH 7.5) containing 5% skim milk powder and 3% bovine serum albumin (BSA), and incubated overnight with primary antibody. Primary antibodies used were: ERα (1:500, MC-20, Santa Cruz; 1:1000 [E115] Abcam ab32063), ERβ (1:750, ab3576, Abcam), PIK3C2B (1:1000, Clone 22/PI3-K, BD Biosciences), DNM2 (1:500 Clone 2862, Dr. Jocelyn Laporte, IGBMC France), DNM2 (1:1000 G-4, Santa Cruz), β-actin (1:5000, #4967 Cell Signaling; 1:5000 Abcam ab8226) and HSP90 (Clone OTI4C10, Origene). After extensive washing in 1X TBST, membranes were incubated for 1 h with horseradish peroxidase-conjugated goat anti-rabbit or goat anti-mouse IgG secondary antibody (1:5000, BioRad) in 1X TBST containing 5% skim milk powder and 3% BSA.

**Densitometry analysis**. Blots were imaged by chemiluminescence (Clarity Max™ ECL, BioRad) using the Gel Doc™ XR + Gel Documentation System (BioRad), and band signal intensities determined using ImageLab software (BioRad). All densitometry values are individually standardized to corresponding values of total β-actin or HSP90 for each probe, and expressed as the fold difference from the average of the WT group of each blot. For each murine drug trial, all samples derived from the same experiment and western blots for each probe were processed in parallel. For tamoxifen cellular drug trials, all samples derived from the same experiment and the samples were run in triplicate, with the exception of the HEK293T, and human fibroblast lines where two to three separate blots were run in parallel. For the MG-132 and Bortezomib cellular drug trials, all samples derived from the same experiment and multiple blots were run in parallel.

**Quantitative real-time PCR**. Total RNA was isolated from quadriceps muscle using the Qiagen RNeasy Kit in accordance with the manufacturer's protocol, and 500 ng was reverse-transcribed using the iScript™ cDNA Synthesis Kit (BioRad). Quantitative real-time PCR (qRT-PCR) for DNM2 was performed using TaqMan Probe assay kits

(ABI; Mm00514582_m1, Mm02619580_g1) in accordance with the manufacturer's protocol, and mRNA levels were normalized to actin as the invariant housekeeping gene. qRT-PCR for ERα was performed using SYBR Green (Thermo Fisher), and mRNA levels were normalized to levels of the TATA box-binding protein (TBP) as the invariant housekeeping gene. All reactions were conducted using the Applied Biosystems StepOne Real-Time PCR System (Thermo Fisher Scientific). Primers used were: 5′-TGCGCAAGTGTTACGAAGTG-3′ (ERα forward), 5′-TTTCGGCCTTCC AAGTCATC-3′ (ERα reverse), 5′-TGCTGCAGTCATCATGAG-3′ (TBP forward), 5′-CTTGCTGCTAGTCTGGATTG-3′ (TBP reverse). The primers used for qRT-PCR were designed in regions that are not affected by alternative splicing from the sequences published under the NCBI Accession numbers NM_007956.4 (mouse ERα), and NM_013684.3 (mouse TATA box-binding protein). mRNA levels were analyzed using the 2$^{-\Delta\Delta CT}$ method as previously described[50].

**Gene mapping and differential expression analysis**. Read sets from each of the 14 samples across the four experimental conditions were aligned to the reference genome (GRCm38/MM10 version of the *Mus musculus* genome) and transcriptome using STAR[51] two-step alignment to generate a Binary Alignment Map file (BAM file). Coordinate sorted BAM files were used to quantify transcript abundance (count data) using HTSeq[52]. Raw read counts generated were used as input for differential gene expression analysis, carried out using both DESeq[53] and edgeR[54] R/Bioconductor packages. This was carried out in a pairwise manner between any two conditions that were considered. FDR adjusted *p*-values from both edgeR and DESeq was used to determine genes that are significantly differentially expressed between the conditions tested. Finally, we used the GoSeq R/Bioconductor package[55] to identify pathways that were significantly enriched for differentially expressed genes between any two conditions.

**Human cell lines**. Human fibroblast lines (f4, f5, f7, and mtm1d) were acquired at the Hospital for Sick Children; mtm1a and mtm1b were purchased from the Coriell Institute (Camden, New Jersey). All lines screened negative for mycoplasma. Lines derived at Hospital for Sick Children were obtained via an REB approved protocol (Dowling) and with patient informed consent (line published in Al-Hashim et al.[56]). Study protocol and consent reviewed and approved by the Hospital for Sick Children Research Ethics Board. Fibroblasts were cultured in media containing DMEM plus 10% fetal bovine serum without addition of antibiotics. For the generation of transdifferentiated myotubes from patient derived fibroblasts, we utilized methodology modified from Fernandez-Fuente et al.[57]. In brief, early passage (<10) fibroblasts were seeded in 10 cm dishes coated with 20% matrigel (Corning) at 50% confluence. At 70% confluence, cells were infected with ad-MyoD (Vector Biolabs) at 125 MOI in infection media (skeletal muscle growth media, Promocel) for 3 h at 37 °C. Media were then changed to differentiation media (DMEM plus 2% horse serum and 0.1% insulin) and cells were incubated until experimental endpoint. Differentiation media were replaced by half every other day. On the 6th day in differentiation media, cells were treated with tamoxifen for 24 h.

**Cell culture and in vitro drug experiments**. MCF-7 cells were cultured and maintained in media containing DMEM and L-glutamine supplemented with 10% fetal bovine serum (FBS) and 100 units/mL of penicillin and 100 μg/mL of streptomycin. MCF-7 cells were obtained from ATCC and were also a gift from Dr. Sean Egan at the Hospital for Sick Children. They tested negative for mycoplasma contamination. Their use for in vitro drug experiments was justified as an estrogen receptor positive cell-line. They were verified for estrogen receptor expression but not otherwise specifically authenticated.

Human fibroblast lines were maintained in media containing DMEM and L-glutamine supplemented with 20% FBS. C2C12 myoblasts and HEK293T cells (ATCC, Virginia, USA) were maintained in media containing DMEM and L-glutamine supplemented with 10% fetal bovine serum (FBS) and 100 units/mL of penicillin and 100 μg/mL of streptomycin. For differentiation to C2C12 myotubes, C2C12 myoblasts at were seeded into six-well plates with a confluence of $5 \times 10^5$ cells/ml. Cells were allowed to reach 90% confluence with 10% FBS before switching to growth media of DMEM and L-glutamine supplemented with 2% horse serum. Fresh media were added daily for 5 days. Cell culture reagents were from Gibco/invitrogen unless otherwise indicated. Twenty-four hours prior to the tamoxifen experiments, cells were transferred to six-well plates at a confluence of $1.5 \times 10^5$ cells/ml. After 24 h, cells were treated with vehicle (0.1% DMSO) or increasing doses of 4H-Tamoxifen (Cayman Chemicals) ranging from 500 nM to 20 μM for 24 h. C2C12 cells and HEK293T were seeded into six-well plates at $2.5 \times 10^6$ cells/ml and $5 \times 10^6$ cells/well respectively, 24 h prior to tamoxifen treatment. Similarly, 1 day prior to proteasome inhibitor treatments, MCF-7 cells were seeded into six-well plates at a confluence of $1.5 \times 10^5$ cells/well. After 24 h, cells were pre-treated for 30 min with proteasome inhibitors 10 μM MG-132 (Cayman Chemicals) or 30 nM Bortezomib (Cayman Chemicals), followed by vehicle (0.1% DMSO) or 10 μM 4H-Tamoxifen for 24 h. For trans-differentiation of primary human fibroblasts to myotubes, all primary human fibroblasts used were early passages (Passage <10). Cells were seeded in 10 cm dishes coated with 20% matrigel (Corning® Matrigel® Matrix) at 50% confluency. Reaching 70% confluency the cells were infected with ad-MyoD (Vector Biolab Cat. No.1492) with a 125 MOI in

infection media (Skeletal muscle cell grow media; Promocel Cat. No.23060). Infected dishes were swirled every 20 min and incubated at 37 °C for 3 h. Media were subsequently changed to differentiation media (DMEM plus 2% Horse Serum plus 0.1% insulin). Differentiation media was replaced by half every other day. On the 6th day of differentiation, the cells were treated with tamoxifen for 24 h as described above[58]. For assessment of β1-integrin localization, HeLa cells were cultured in DMEM 4.5 g/L glucose (Lonza) supplemented with 0.1 units/mL of penicillin, 0.1 μg/mL of streptomycin and 10% fetal calf serum (Gibco). HeLa cells were depleted of endogenous MTM1 by double transfection on days 1 and 3 with siRNA targeting MTM1 (5′-GATGCAAGACCCAGCGTAA-3′) or control siRNA (5′-AAATCGGATATCGGAATAG-3′) (Dharmacon) using Oligofectamin (Life Technologies). On day 4 cells were seeded onto Matrigel-coated coverslips and treated with 1 μM Tamoxifen (Sigma-Aldrich) or ethanol (vehicle) for 12 or 24 h before fixation with 4% PFA for 10 min. HeLa cells were co-stained with antibodies against β1-integrin (clone LM534, Millipore) and transferrin receptor (TfR; rabbit-anti-TfR, Sigma_Aldrich) as previously described[4]. Confocal images were acquired on a Zeiss CSU spinning disk microscope and analysed with ImageJ software. Three independent experiments were performed and between 72 and 100 cells were analysed per condition in each experiment. The data were statistically analysed by One-way ANOVA using SigmaPlot software. All cells used in in vitro studies were tested for mycoplasma contamination.

**Data and statistical analysis**. Unless otherwise specified, all data presented are expressed as the as the mean ± SEM. For all mice experiments, all $n$ values represent discrete individual mice. With the exception of RNA sequencing data, postcapture analysis was performed using Microsoft Excel (2008). GraphPad Prism software, version 7.0 for MacOSX (GraphPad) was used for constructing all graphs and for performing all statistical analyses. Differences between groups were assessed by ordinary two-way ANOVA accounting for multiple comparisons. Posttest analysis utilized either Tukey's multiple comparisons or Fisher's Least Common Differences. For direct two sample comparisons, unpaired, parametric two-tailed Student's $t$-test was performed. For survival curves, the log-rank (Mantel-Cox) test was performed. Uncropped scans of western blots supporting data in this study are available separately as a Supplementary Information file.

**Primers**. Mouse ERα forward 5′-TGCGCAAGTGTTACGAAGTG-3′
Mouse ERα reverse 5′-TTTCGGCCTTCCAAGTCATC-3′
Mouse TBP forward 5′-TGCTGCAGTCATCATGAG-3′
Mouse TBP reverse 5′-CTTGCTGCTAGTCTGGATTG-3′
siRNA targeting MTM1—5′-GATGCAAGACCCAGCGTAA-3′
control siRNA—5′-AAATCGGATATCGGAATAG-3′

**Reporting Summary**. Further information on research design is available in the Nature Research Reporting Summary linked to this article.

## Data availability

RNA sequencing data supporting this study has been deposited into the Gene Expression Omnibus (GEO) database (NCBI) and is accessible under GEO Series accession number GSE120336.
All raw data points, pathology and microscopy images, and uncropped western blots supporting the findings in this study are available from the corresponding author upon request. A reporting summary for this article is available as a Supplementary Information File.

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

## Acknowledgements

The authors thank Ronald Cohn and Sean Egan for critical reading of the manuscript, Hernan Gonorazky for assistance with blinded quantification, and Jonathan Volpatti for help with figure generation. This work was supported by grants from the Myotubular Trust (17SIC01, J.J.D.), Canadian Institute of Health Research (324830 and 376691, J.J.D. plus subcontract to R.T.D.), and National Science and Engineering Research Council (NSERC) (RGPIN-2015–05322, J.J.D.). It was also supported by the Mogford Campbell Family Chair fund (J.J.D.).

## Author contributions

J.J.D. conceived the project. J.J.D., N.M., and N.S. designed and conducted the majority of all experiments and analysed the data. J.J.D. wrote the manuscript with input from all coauthors. N.M., N.S., and J.J.D. edited and corrected the manuscript. K.R. performed in vitro TAM screens and the PIK3C2B kinase assay. A.R. and M.B. compiled and conducted an analysis of all bioinformatical data pertaining to RNA sequencing. G.R. and V.H. performed the β1-integrin localization studies. L.G. and R.T.D. performed EC coupling analyses and interpretation. N.E., F.M., and A.P. performed aspects of the experimentation and data analysis.

## Additional information

**Competing interests:** The authors declare no competing interests.

