## [Peer Review File · Nature Communications]

Response to Reviewers

We thank the reviewers for the thorough reading of the manuscript, as well as the helpful comments and critiques.

Reviewer #1:

In the present paper Maani and colleagues present convincing evidence for a beneficial effect of Tamoxifen in a well-established mouse model of X-linked myotubular myopathy (XLMTM), one of the most devastating inherited neuromuscular conditions in humans, and a detailed and plausible explanation of the underlying molecular mechanisms. This is an important and timely study concerning a profoundly severe neuromuscular condition for which there is currently no cure, from a group with a proven track record in elucidating mechanisms of early-onset neuromuscular disorders with potential relevance for clinical translation.

We thank reviewer 1 for the overall positive evaluation of the manuscript

I have few comments and queries:

*Page 4: Tamoxifen treatment was started initially at 21 days of age which is reasonable, considering that this is the point in time where *mtm1* KO mice usually become symptomatic in terms of decreased body weight but still have preserved muscle strength. Considering that the disease process is likely to start before clinical disease manifestations become obvious, would the authors expect an even further increase in lifespan if Tamoxifen treatment was commenced immediately postnatally, based on genotype rather than phenotype?*

We thank reviewer 1 for this excellent point. To address it, we sought to treat animals at an earlier time point. We chose to treat at 14 days postnatal and with low dose tamoxifen. This was based on discussions with our animal facility, and in talking with other investigators that use tamoxifen frequently in mice, after which we surmised that (a) high dose treatment is not tolerated by feeding mothers and (b) 1 week of low dose treatment (14-21 days) is safe and tolerated by breastfeeding mothers and their pups. *Mtm1* KO mice at 14 days of age are not distinguishable for their wild type siblings based on histopathology and muscle strength, but may be smaller in overall appearance.

Chronic daily treatment of low dose TAM starting at 14 days had a dramatic effect on phenotype and survival. KO animals appeared normal at all time points up until around 55-60 days, as reflected by normal quantitative grip strength at 35 and 50 days. In contrast, untreated *Mtm1* KOs have significant weakness at 35 days and are typically dead at 50 days. *Mtm1* KOs treated starting at 21 days have normal grip at 35 days but have advanced weakness at 50 days to the point that they are not able to perform the grip test at this stage.

Furthermore, survival of mice treated at 14 days is consistently extended beyond 70 days (median survival 71 days, longest survival 83 days, n = 9), reflecting a statistically significant improvement in survival with this early treatment as compared to no treatment or with treatment starting at 21 days.

These data are now included in Figure 1B and in a new Supplemental figure (Supplemental Figure 2), and are additionally demonstrated with Supplemental video 5.

*Page 5: Three of the *mtm1* KO mice in the subgroup treated with high dose Tamoxifen died within the first 2-3 days of therapy onset. I am not fully convinced that this can be entirely explained by reduced food intake and consequent weight loss as suggested in the next paragraph – have other causes of death in these animals be carefully excluded? Although Tamoxifen is a well-established drug with a*

good safety profile, considering that myotubularin deficiency may be associated with non-skeletal muscle manifestations in humans, it must be ascertained that these unexpected deaths do not reflect a particular susceptibility of the myotubularin-deficient state to Tamoxifen-related side effects.

We have further investigated cause of death in *Mtm1* KO animals treated with high dose tamoxifen. In the small subset of animals that die within the first 1-3 days of therapy initiated at 30 days of age, in addition to skeletal muscle pathology, we have detected pulmonary hemorrhage that is likely the cause of death. We did not observe pulmonary hemorrhage in untreated *Mtm1* KO mice at end point, or in any other treatment group (low dose starting at 21 or 30 days, high dose starting at 21 days). We also did not see it in the longer-lived MTM mice started on TAM at 30 days. It thus appears restricted to this small subset of late stage, high dose treated MTM mice. We report these findings in Supplemental figure 3, and comment on them as well in the discussion section.

Of note, there are a few case reports of acute pulmonary injury in breast cancer patients receiving tamoxifen, and pulmonary toxicity is a recognized very low frequency side effect of the drug. Importantly, this is seen in our experiment only in a few late stage, high dose treated MTMs, and not with our low dose treatment. Our low dose treatment is more reflective of paediatric dosing in humans, and it promotes similar survival and motor functional benefit as high dose in our MTM mice. We therefore feel this observation does not impact the translatability of our findings; nevertheless, pulmonary monitoring should likely be considered as part of adverse event surveillance for any clinical trial of tamoxifen for MTM.

Page 6: Do the authors want to speculate on the molecular basis for the observed fiber type changes, with an increase in type 2B and reduced type 1 and 2A fibers?

We hypothesize that the fiber type changes are due to improvements in intracellular calcium handling and restoration of the EC coupling apparatus (as shown in Figure 3). There is support for this idea in the biomedical literature. For example, restoring DHPR expression in a mouse model of myotonic dystrophy leads to a shift from type 1 to type 2B fibers. We briefly mention this possible explanation on page 7, last paragraph.

Page 9, Line 190: “extra-membrane signalling” – do you mean “extra-nuclear signalling”?

Yes, extra-nuclear. We have corrected this.

Discussion

*Considering the observed ability of Tamoxifen to reduce DNM2 in both *mtm1* KO and wt muscles, do the authors want to speculate if Tamoxifen may also be a feasible option in other forms of CNM where DNM2 modulation has been considered as a therapeutic modality, for example BIN1-related CNM?*

This is an excellent point. We do think that tamoxifen should be considered as a potential therapeutic for other forms of CNM, particularly given that DNM2 knockdown has been shown to ameliorate aspects of the phenotype of a *Bin1* KO mouse, a model of another form of CNM. At present, it is not feasible to test tamoxifen in other models of CNM (the *Bin1* KO mice mentioned above die at birth, existing mice that model DNM2-CNM have no overt phenotype, and there is no mouse model of RYR1 related CNM). We are working to develop models suitable for drug development, and plan to test tamoxifen in them once they are established. We have now added mention of this possibility to the discussion section.

Figure 3b: Although this may not be statistically significant, at least on visual inspection the number of triads in wt mice treated with high dose TAM appears to be reduced compared to the number of triads in untreated wt mice – can the author comment on this observation?

There does appear to be a trend toward decreased number of triads, though the magnitude of change is small and not statistically significant. While this is potentially not a meaningful trend, one possibility is that the reduction in DNM2 below normal levels seen in high dose WT animals is associated with a slight reduction triad number. Alternatively, given the small shift of fiber type, it is possible that type IIB fibers have slightly fewer triads than Type I or Type IIA.

Reviewer #2

This manuscript by Maani et al gives substantial evidence that tamoxifen treatment ameliorates the phenotype of a mouse model of X-linked myotubular myopathy, MTM1. The authors used two doses of tamoxifen 3mg/Kg and 40mg/kg. The lower dose increased survival of the Mtm1 mice to 57 days and the high dose to 48 days from 39 days. There was improvement in grip strength, improvement in muscle histopathology, increase in calcium transients. Estradiol, an estrogen receptor agonist increased survival, whereas Fulvestrant and estrogen receptor antagonist did not, suggesting the tamoxifen effect was at least in part related to estrogen receptor agonism. There was reduction in the level of DNM2 protein. Reduction of DNM2 is known to be beneficial in mouse models of MTM1. The authors therefore conclude that some of the tamoxifen effect may be through reducing DNM2 levels. The distribution of ERα receptors is abnormal in Mtm1 KO mice and this was at least partially normalised by the tamoxifen treatment. This too may be part of the mechanism of therapeutic benefit. Finally the authors demonstrate that tamoxifen effect may in part occur through activation of the ubiquitin-proteasome system. This is a comprehensive study of multiple possible mechanisms through which tamoxifen may be producing benefit to a mouse model of MTM1. Tamoxifen is a drug that is in use in paediatric patients. It may therefore be trialled in MTM1 patients relatively easily. This is a result that should be of significant interest to the muscle disease field and also outside the field

We thank reviewer 2 for the thorough review and for the positive comments related to the study and its potential impact.

General comments.

I would suggest the authors remove all use of the word “first” or similar wording from the manuscript. It is common practice these days not to claim “firsts” and this is especially relevant here because of the jointly submitted manuscript.

We have removed such mentions.

In many of the figures the font size is too small to be legible on the printed page.

We have corrected the font size on the figures to make them more comprehensible

The authors use “expression” when referring to protein levels. I have suggested replacing “expression” with “level” wherever I have noticed this. The authors should do this throughout when “expression” refers to protein levels.

We have changed “expression” to “level” where applicable

The manuscript is mostly clear. Some of the figures used do not appear to be the correct figures or are not adequately explained.

We have made sure all figures correlate appropriately with the text and have provided a thorough explanation for each.

Specific comments:

Page 4 of the PDF lines 77-78: the low dose tamoxifen increased survival more than the high dose tamoxifen. The authors suggest that the problem is due to the reduction in food intake and associated weight loss with high dose tamoxifen. The jointly submitted manuscript by Dorchie's group does not see such an effect of their high dose tamoxifen. Do the authors have any explanation for the difference?

In the jointly submitted manuscript by the Dorchie's group, their “high dose” treatment is equivalent in dosage to our “low dose”. We do not see weight loss with low dose treatment (in keeping with their high dose data).

On page 6 of the PDF (lines 106-108) low and high dose tamoxifen are shown to restore grip strength. On page 8 of the PDF, (lines 164-166), the statement is made that both low and high dose tamoxifen as well as estradiol, reduced the amount of ERα protein. In the discussion, (Page 14 of the PDF, lines 295-297), it is stated that reduction of estradiol levels in females and knockout of ERα in mice lead to impaired muscle force. These seem contradictory statements. Would the authors please comment?

We agree that these statements seem potentially contradictory. However, there are important caveats to consider with the previous study vis-à-vis our findings. Firstly, the *Era* KO study only examined female mice, and thus we do not know whether there is any impact on genetically lowering ERA in male mouse muscle. Secondly, the *Era* KO study found weakness in mice only at 26 weeks and beyond, time points well beyond those examined in our study. Thirdly, ERA levels in untreated *Mtm1* KO mice are significantly elevated, and TAM treatment reduces this elevation to levels that approximate wild types.

Minor questions:

Supplemental Figure 1: the body weights of the untreated control WT mice are very different between the high dose and low dose treatment graphs, with the body weight of the WT untreated control mice for the high dose TAM mice continuing to increase up towards 30g but plateauing in the low dose TAM graph around 20g. Do the authors have an explanation for this difference?

We thank reviewer 2 for pointing this out. We have now included the full set of untreated WT mouse littermates in the weight calculation for the low dose treatment category, and the weights match those of untreated WT in the high dose group.

Minor comments

In the life sciences reporting summary, under the heading “Blinding”, the meaning of the wording: “The slides were only identifiable by code (set by the two initial individuals) related to mouse ID number),” is unclear.

Two individuals collected the samples and gave them anonymized/coded numbers. A third individual then evaluated the samples. This individual was blinded to genotype and treatment, and scored all parameters and linked them to the coded number.

Page 5 of the PDF: The Supplemental videos need greater explanation of what they are showing. What are all the mice in each video? For example, in the “high TAM 29 days” video is there an untreated KO, a wt and a high-TAM treated KO, or what? In the “low TAM 68-days” video a note tells us there is a WT TAM-treated mouse and a KO TAM treated mouse. You can work out that the mouse that moves up the left-hand side of the cage is probably the treated KO, because the back legs don’t work normally. But it would be good to have the mice labelled somehow, or an explanation given.

We have now included labels in the videos so that each of the genotypes is more easily identified.

Page 6 of the PDF – please comment on the subsarcolemmal accumulations seen in the SDH staining. What are they? Are they ameliorated by the treatments?

The subsarcolemmal accumulations have been examined at the ultrastructural level in previous studies and are found to be primarily mislocalized mitochondria and sarcoplasmic reticulum (and include as well other organelles). They have been termed “necklace fibres” when seen in MTM patient biopsies. They were originally identified in milder cases of MTM and in female carriers, though more recently have also been seen in patients with classic, severe MTM. They seem to be more frequently seen in biopsies from older individuals, and so may be an age related pathologic phenomenon. Their exact significance is unknown.

Page 9 of the PDF, lines 189-190: Why do the ER α localisation results suggest a role for extra-membrane signaling? That is perhaps going beyond the data. Perhaps simply state that signaling at the sarcolemma may be involved.

We do believe that our transcriptome data and the elucidation of ER α localization at the sarcolemmal membrane lends support to a hypothesis that ER α is functioning via “extra nuclear” signaling pathways in this setting. However, we agree that this assumption goes a bit beyond our data, and thus have modified the statement to say “these results suggest that extra-nuclear signaling may be involved as a potential mechanism for ER α and tamoxifen in MTM”.

Page 10 of the PDF, lines 204-206: “Tamoxifen treatment of HeLa cells, either with or without Mtm1 siRNA, did not alter PIK3C2B signaling (Supplemental Figure 6).” Supplemental Figure 6 shows results of tamoxifen treatment on stalling of endosomes. Is this what Supplemental Figure 6 is supposed to show?

We apologize for the confusing description of these data. As the reviewer correctly states, we have in this figure tested whether tamoxifen can reverse the stalling of endosomal exocytosis seen with knockdown of *Mtm1*. This defect is felt to be PI3P dependent, and can be rescued by knockdown of either *Pik3c2b* or *Vps34*. Tamoxifen, however, is not able to reverse it. We believe this provides additional supporting evidence that tamoxifen is not acting via modulation of PI3P and class II/III PI3 kinase inhibition. We have modified the text to make this point clearer.

Page 11 of the PDF, lines 219-220: “No treatment changed Dnm2 transcript levels (Supplemental

Figure 7)”. *Supplemental Figure 7 is labelled as showing MTM1 expression (though it is a western blot) in primary human fibroblasts, not Dnm2 transcript levels. Is this the correct Figure reference?*

The correct figure is Supplemental Figure 9. We apologize for the mislabeling.

Page 11 of the PDF, line 232: remove “expression”. Page 12 of the PDF, line 240 replace expression with “level”. Figure 2 legend: “(3) resolution of mitochondrial aggregation” from the figure cannot really say “resolution”, perhaps better to say “reduction”. Figure 6: Title should be: “Tamoxifen reduces dynamin-2 protein levels in vivo and in vitro” not “Tamoxifen reduces dynamin-2 protein expression in vivo and in vitro” since protein level not mRNA level is being measured. Supplemental Table 2 Title: rather than: “Spreadsheet of transcripts significantly enriched with comparative RNA sequencing” Suggest: “Spreadsheet of transcripts significantly enriched in comparative RNA sequencing” Supplemental Table 4 title – remove “from” Supplemental table 7 remove “in” from “levels in under various” Supplemental Figure 8. Title should be: “Dose-dependent effects of in-vitro tamoxifen treatment on dynamin-2 protein levels in human fibroblasts.” Supplemental Figure 10 title should be. “Levels of estrogen receptor alpha in C2C12 myotubes and HEK293T cells.” Materials and Methods, Page 8 line 184: “For assessment of __-integrin localization” the squares in the PDF need resolved.

We appreciate the reviewer identifying and pointing out these changes. We have made all of the relevant corrections.

Reviewer #3:

The authors are presenting an interesting work demonstrating efficacy of tamoxifen in a preclinical mouse model of Myotubular myopathy (MTM) that is a severe X-linked congenital myopathy affecting in most cases infants with unmet needs for treatment. The authors show interesting data in the MTM mouse demonstrating a clear increase in survival, and they also nicely demonstrate how the beneficial effects of tamoxifen are mediated primarily via estrogen alpha receptor signalling with post transcriptional reduction of dynamin-2. The results of this team show preclinical evidence that tamoxifen can be considered a repository molecule to hasten clinical trials in patients affected by congenital myopathy related to the X-linked MTM1.

We thank reviewer 3 for the evaluation of our manuscript and the comments related to our findings and their implications.

Overall the results of this study overlap with the other back to back work presented by Dorchies et al. However, a point that does not seem to coincide with the study of Dorchies et al. is that tamoxifen does not seem to downregulate PIK3C2B in the study by Dowling et al., while PIK3C2B is found to be high compared to controls in their transcriptomic analysis and normalize after treatment with tamoxifen in the work by Dorchies et al. that worked in the same animal KO mouse model. Can the authors clarify this point, can they comment on the transcriptomic analysis and clearly state about the levels of expression of PIK3C2B in the muscle of KO MTM mouse compared to controls?

We have repeated our original measurements of PIK3C2B, and additionally looked at a second muscle group. We continue to see no difference in levels of PIK3C2B. The reason for the difference in PIK3C2B expression between our study and the Dorchies study is unclear. One potential explanation is that the mice are maintained on different genetic backgrounds (C57BL6 in our study, 129/Agouti in the Dorchies study), and that PIK3C2B levels may differ because of it.